# Shortwave and longwave components of the surface radiation budget measured at the Thule High Arctic Atmospheric Observatory, Northern Greenland

Daniela Meloni[1], Filippo Calì Quaglia[2,3], Virginia Ciardini[1], Annalisa Di Bernardino[4], Tatiana Di Iorio[1],
Antonio Iaccarino[1], Giovanni Muscari[3], Giandomenico Pace[1], Claudio Scarchilli[1], and Alcide di Sarra[5]

[1]Laboratory for Observations and Measurements for Environment and Climate, ENEA, Rome, 00123, Italy
[2] Department of Environmental Sciences, Informatics and Statistics, Ca' Foscari University of Venice, Mestre, 30172, Italy
[3]INGV, Rome, 00143, Italy
[4]Physics Department, Sapienza University of Rome, Rome, 00185, Italy
[5]Laboratory for Observations and Measurements for Environment and Climate, ENEA, Frascati, 00044, Italy

*Correspondence to*: Daniela Meloni (daniela.meloni@enea.it)

**Abstract.** The Arctic climate is influenced by the interaction of shortwave (SW) and longwave (LW) radiation with the atmosphere and the surface. The comprehensive evolution of the Surface Radiative Fluxes (SRF) on different time scales is of paramount importance to understanding the complex mechanisms governing the Arctic climate. However, only a few sites located in the Arctic region provide long-term time series of SRF allowing for capturing the seasonality of atmospheric and surface parameters and carrying out validation of satellite products and/or reanalyses.

This paper presents the daily and monthly SRF record collected at the Thule High Arctic Atmospheric Observatory (THAAO, 76.5° N, 68.8° W), in North-Western Greenland. The downwelling components of the SW (DSI) and the LW (DLI) irradiances have been measured at THAAO since 2009, while the collection of the upwelling SW (USI) and LW (ULI) irradiance was started in 2016, together with additional measurements, such as e.g., meteorological parameters and column water vapour. The datasets of DSI (Meloni et al., 2022a; https://doi.org/10.13127/thaao/dsi), USI (Meloni et al., 2022b; https://doi.org/10.13127/thaao/usi), DLI (Meloni et al., 2022c; https://doi.org/10.13127/thaao/dli), ULI (Meloni et al., 2022d; https://doi.org/10.13127/thaao/uli), and near surface air temperature (Muscari et al., 2018; https://doi.org/10.13127/thaao/met), can be accessed through the THAAO web site (https://www.thuleatmos-it.it/data).

DSI is absent (solar zenith angle≥90°) from 29 October to 13 February, assuming maxima in June (monthly mean of 277.0 Wm$^{-2}$), when it is about half of the total incoming irradiance. The USI maximum occurs in May (132.4 Wm$^{-2}$) due to the combination of moderately high DSI values and high albedo. The shortwave surface albedo (A) assumes an average of 0.16 during summer, when the surface is free of snow. Differently, during months of snow-covered surface, when solar radiation allows estimating A, its values are greater than 0.6. A large interannual variability is observed in May and September, months characterized by rapidly changing surface conditions, having a link with air temperature anomalies.

DLI and ULI maxima occur in July and August, and minima in February and March. ULI is always larger than DLI and shows a wider annual cycle. ULI is well described by a fourth-order polynomial fit to the air temperature ($R^2>0.99$ for monthly data and $R^2>0.97$ for daily data).

The surface radiation budget (SRB) is positive from April to August, when absorption of solar radiation exceeds the infrared net cooling, with a maximum value of 153.2 Wm$^{-2}$ in June. From November to February, during the polar night, the LW net flux varies between -34.5 and -43.0 Wm$^{-2}$. In March and September, the negative LW net flux overcomes the positive SW contribution, producing a negative SRB.

THAAO measurements show clear evidence of the influence of several regional weather/climate events, that appear strongly
linked with SRF anomalies. These anomalies are found for example during summer 2012, when a large ice melting event took place over Greenland, and during winter 2019-2020, extraordinarily cold in the Arctic region.

## 1 Introduction

Solar and infrared radiation are key elements of the Arctic Amplification, that is the result of complex interactions involving the atmosphere, the cryosphere, the land and the oceans (Serreze and Francis, 2006). Due to its complexity, the Arctic
Amplification is widely studied by applying climate models that take into account many different factors, such as increased carbon dioxide and temperatures, variations in surface albedo, water vapour, and clouds, linked by feedback mechanisms, modifying the radiation flux at the top of the atmosphere (TOA) and at the surface (Goosse et al., 2018; Dai, 2021). However, significant differences among climate models in simulating Arctic warming have been observed. These can be attributed, among other causes, to difficulties in the representation of radiative processes and forcings (Bintanja and Krikken,
2016), often simplified by parameterizations that have to be verified and, hopefully, improved employing ground-based measurements.

The major feedback mechanisms driving Arctic climate involve radiation and its interactions with clouds and the surface, such as the ice-albedo and the cloud-radiation feedback (Curry et al, 1996; Taylor et al., 2013). In the ice-albedo feedback the decrease of surface albedo associated with sea ice reduction causes an additional absorption of solar radiation which, in
turn, produces an enhanced sea ice melting. The projected increase in low clouds during autumn and winter, with a minimum insolation, shall enhance the lower atmosphere emissivity and the downwelling longwave radiation, contributing to additional surface warming and sea ice melting (Previdi et al., 2021).

The polar climate is influenced by radiative processes related to clouds, resulting from the balance of shortwave cooling and longwave heating, critically depending on the clouds' physical and optical properties, such as the cloud phase and the cloud
optical depth (e.g., Kay et al., 2016; Ebell et al., 2020). Changes in surface albedo affect cloud properties, largely influencing shortwave and longwave radiation (e.g., Kapsch et al., 2016; He et al., 2019).

Understanding the complexity of the Arctic climate requires a comprehensive evaluation of the Surface Radiative Fluxes (SRF) on different time scales, to understand the link with rapidly varying atmospheric components (mainly clouds and

aerosols) and to detect slow variations associated with long-term changes in climatic parameters. Surface radiative fluxes can be derived from ground-based measurements of the incoming and outgoing shortwave (SW) and longwave (LW) irradiances, as for example in sites belonging to the Baseline Surface Radiation Network (BSRN) (Ohmura et al., 1998; Driemel et al., 2018), but also from satellite observations (e.g., Stackhouse et al., 2011; Rutan et al., 2015; Karlsson et al., 2017), and reanalyses (e.g., Rienecker et al., 2011; Dee et al., 2011).

Most studies on Arctic warming rely on atmospheric reanalyses data that, however, have shown to be biased in the Arctic surface temperature (e.g., Orsi et al., 2017; Batrak and Müller, 2019) as well as in the simulation of persistent summer clouds, leading to errors in the estimation of SRF (Graham et al., 2019). The limitation in ground-based observations and in their spatial distribution is the main factor that does not allow a proper constrain of model reanalysis (Batrak et al., 2023).

Although high latitudes benefit from the frequent passages of satellite sensors, retrieving surface radiation from outgoing radiance measurements at the TOA may introduce errors associated with the atmospheric composition and/or with the vertical distribution assumed in the computation (Bourassa et al., 2013). Moreover, frequent high cloud cover conditions further complicate the retrieval of satellite data at polar latitudes. For these reasons, surface radiation fluxes derived from satellite observations require a comprehensive validation against ground-based measurements to assess their quality, particularly in the polar regions, where biases have been detected mostly due to misrepresentation of clouds and/or surface properties (Riihelä et al., 2017; Blanchard et al., 2021; Di Biagio et al., 2021; Wang et al, 2021; Huang et al., 2022).

Unfortunately, only few sites located in the Arctic region provide time series of SRF long enough to conduct systematic comparisons with the SW and LW components of the SRF, and to capture the seasonality of atmospheric and surface parameters. The harsh climatic conditions make it difficult to maintain long-term sites collecting SRF measurements with adequate accuracy to validate satellite or reanalysis products. One of the primary problems is related to the deposition of snow/ice on the radiometers' dome, notwithstanding the active ventilation (Cox et al., 2021) in stations with a discontinuous presence of personnel.

The Arctic stations with long-term records of downward and upward SRF and atmospheric measurements are presented in Matsui et al. (2012). Utqiaġvik, formerly known as Barrow (71.3° N, 156.6° W), Alaska (Dong et al., 2010), and Ny-Ålesund (78.9° N, 11.9° E), Svalbard (Maturilli et al., 2015), have the longest records of SRF, starting in 1976 and 1988, respectively. SRF measurements at Summit (72.68° N, 38.58° W), Greenland (Miller et al., 2015; 2017), began in 2004. Alert (82.47° N, 62.5° W) and Eureka (80.05° N, 86.42° W), Nunavut, Canada, started in 2004 and 2007, respectively. Finally, Tiksi (71.6° N, 128.9° E), East Siberia, has been operating SRF measurements since 2010 (Grachev et al., 2018). All these sites contribute or contributed in the past to the BSRN database.

The importance of combining data from different observatories in the Arctic is highlighted by Uttal et al. (2016). Under the umbrella of the International Arctic Systems for Observing the Atmosphere (IASOA), measurements of the Arctic observatories are used to give an integrated perspective of the regional climate, coping with the diversity of geographical and climatic conditions.

Shorter records have been collected during intensive field campaigns, such as the Surface Heat Budget of the Arctic (SHEBA), in the Beaufort Sea, from October 1997 to October 1998  (Uttal et al. 2002), the Arctic Summer Cloud Ocean Study (ASCOS), in the Arctic Ocean in late summer 2008 (Tjernström et al., 2014), the Norwegian Young Sea Ice Cruise (N-ICE2015) campaign over sea ice north of Svalbard (80–83° N, 5–25° E) from January to June 2015 (Walden et al., 2017), the joint Arctic Cloud Observations Using Airborne Measurements during Polar Day (ACLOUD) campaign and Physical Feedbacks of Arctic Boundary Layer, Sea Ice, Cloud and Aerosol (PASCAL) ice breaker expedition around Svalbard, Norway, in May and June 2017 (Wendisch et al., 2019) and the Multidisciplinary drifting Observatory for the Study of Arctic Climate (MOSAiC) expedition from October 2019 to September 2020 (Shupe et al., 2022). The measurements collected during MOSAiC were used to evaluate the performance of state-of-the-art and experimental forecast models in predicting short-term surface energy fluxes in wintertime. One of the main findings is that biases in the simulated longwave irradiance components are found against surface observations, caused by difficulties of the models in representing liquid-bearing clouds at cold temperatures (Solomon et al., 2023). Modelling studies based on SHEBA measurements had already identified wintertime biases in regional climate models (e.g., Wyser et al., 2008) and in global forecast systems (e.g., Simjanovski et al., 2011). In both cases surface albedo and cloud representations, with their radiative effects, are the main reasons for the estimated biases.

This paper presents the SRF record collected at the Thule High Arctic Atmospheric Observatory (THAAO, 76.5° N, 68.8° W), in North-Western Greenland, located within the Pituffik Space Base (formerly Thule Air Base) area. The Observatory has a long history of measurements of atmospheric composition and upper air vertical structure, starting in 1990 (di Sarra et al., 1992; Larsen et al., 1994; Rosen et al., 1997; Muscari et al. 2007). Recently, great efforts have been devoted to monitoring and studying Arctic troposphere, surface properties, and radiation budget (see Section 2.1 for details).

The downwelling components of the SW (DSI) and LW (DLI) irradiances have been measured at THAAO since 2009. Since 2016, when the upwelling SW (USI) and LW (ULI) irradiance measurements started, the Surface Radiation Budget (SRB) can be derived and analyzed; in parallel, additional measurements began, related to DLI and ULI, such as meteorological parameters and column water vapour.

The availability of THAAO long-term SRF measurements may add knowledge of the Arctic climate in relation to the interaction of SW and LW radiation with aerosol and clouds, thus supporting the improvement of related processes in regional and global models. In addition, radiation measurements at THAAO are valuable for assessing and validating satellite products.

The instruments and the methodologies to obtain irradiances by applying calibrations and corrections to the raw data are discussed in Section 2. A first description of the overall evolution of SRF and SRB at THAAO, also in relation to known regional seasonal anomalies, is given in Section 3. The information on the data availability is in Section 4, while conclusions are discussed in Section 5.

## 2 Instruments

### 2.1 Ground-based instruments at THAAO

The Thule High Arctic Atmospheric Observatory (THAAO, 76.5° N, 68.8° W, 220 m amsl) was set up in the 1990s with a collaborative effort of Italian and Danish institutions: the Danish Meteorological Institute (DMI), the University of Rome "Sapienza", and the Italian National Agency for New Technologies, Energy, and Sustainable Economic Development (ENEA). In 1999, the National Center for Atmospheric Research (NCAR) joined the collaboration followed by the Istituto Nazionale di Geofisica e Vulcanologia (INGV) in 2009. The collaboration between DMI, at that time already involved with measures dedicated to the study of the Arctic climate at the Thule Air Base, and the Italian institutions started with the installation of an aerosol/Rayleigh lidar by the University of Rome in addition to the DMI instruments (ozonesondes and UV/visible spectrometer), to improve knowledge of the stratospheric ozone depletion phenomenon that was observed to be very intense over Antarctica and anticipated to become important also over the Arctic. (Muscari et al., 2014). In 2017, DMI ceased its activities at THAAO, and the Observatory is now managed by the US National Science Foundation (NSF).

Since 2010 THAAO is located in building #1971, on a 220 m high hill South of the base (Figure 1). The site is about 3 km South-East of the coastline, and just South of the terminus of the Wolstenholme Fjord, about 20 km West of the ice sheet.

The Observatory contributes to the Network for the Detection of Atmospheric Composition Change (NDACC, https://ndacc.larc.nasa.gov/) with lidar (Di Biagio et al., 2010) and Fourier transform infrared spectroscopy (FTIR) measurements (Hannigan et al., 2009).

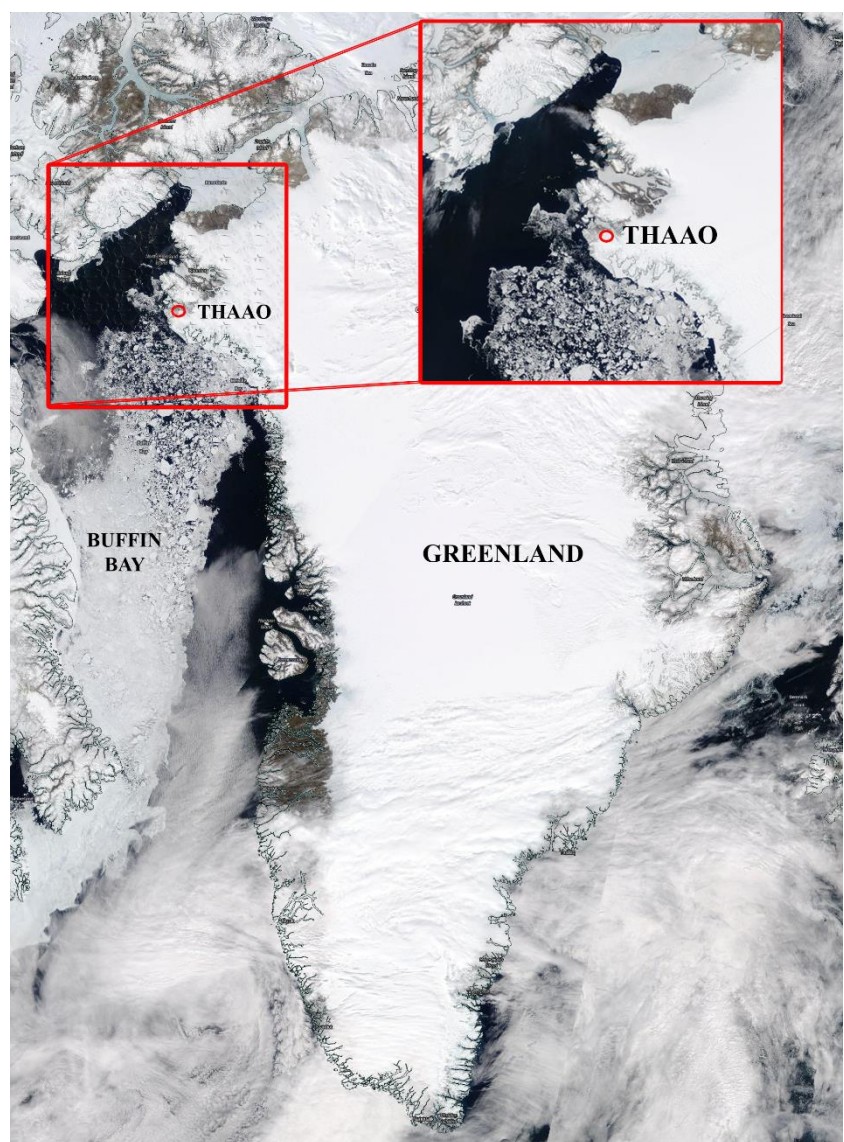

Figure 1. Position of THAAO (Credits: NASA EOSDIS Worldview, https://worldview.earthdata.nasa.gov/).

The first radiometer, a Yankee Environmental Systems Inc. TSP-700 pyranometer, was installed in 2003 by DMI and operated until 2012 (Di Biagio et al., 2012). In February 2009, permanent and continuous measurements of downward longwave irradiance (DLI) and downward shortwave irradiance (DSI) were started with the installation of an Eppley

Precision Infrared Pyranometer (PIR) and a Precision Spectral Pyranometer (PSP). Various PSPs and PIRs, as well as Kipp&Zonen pyranometer model CMP21 and pyrgeometer model CGR4, have been operated at THAAO throughout the years (Figure 2).

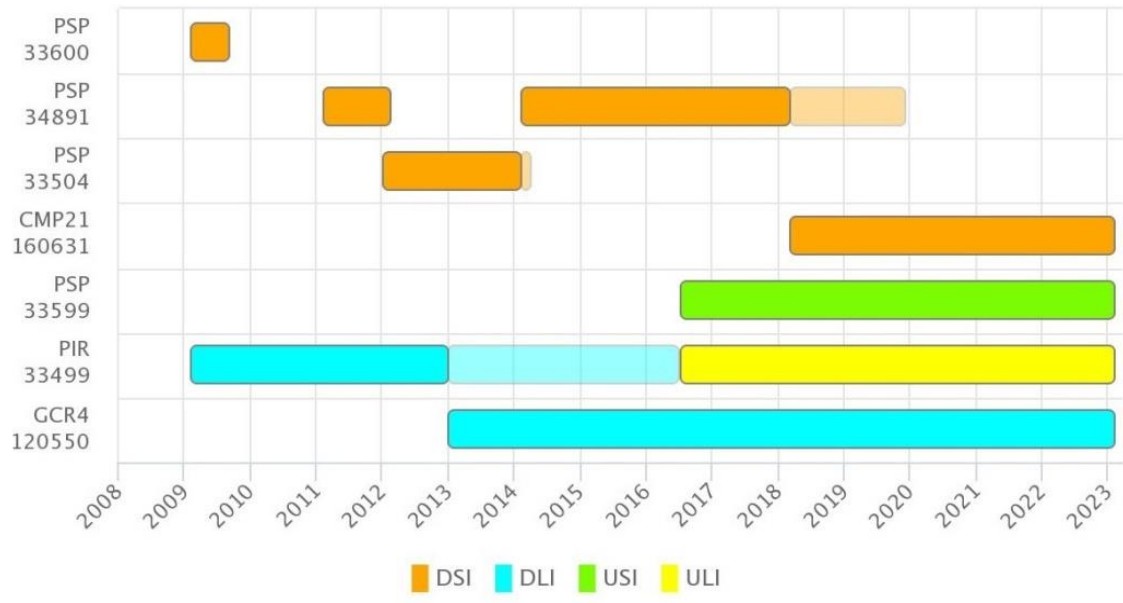

Figure 2: Radiometers installed at THAAO in the whole period 2009-2022. Shaded areas represent periods when a radiometer was tested by comparison with one having a fresh calibration.

Radiometers were installed on the roof of building #1985 before September 2012, and on the roof of building #1971 afterwards. Building #1985 was located about 700 m West of #1971, at about the same altitude. At both sites, the radiometers' horizon is free from obstacles. The radiometers are ventilated to prevent rain/snow/ice deposition. The altitude of the radiometers is about 5 m above ground (Figure 3a). Data from the two sites have been included in the same dataset, due to the small distance between the two sites and negligible altitude difference.

Downward-looking PIR and PSP were installed in July 2016 on a pre-existing metal frame about 50 m South-West of building #1971 to continuously measure the upward longwave (ULI) and shortwave (USI) irradiances. These instruments, placed approximately 2.3 m above ground level, are not ventilated (Figure 3b). The radiometers are mounted on a plate, that extends roughly southward in order to minimize the influence of the supporting frame.

In addition to the SW and LW irradiances, also the downwelling and upwelling photosynthetically active radiation (PAR) has been measured since 2016 with Licor Li-190R sensors.

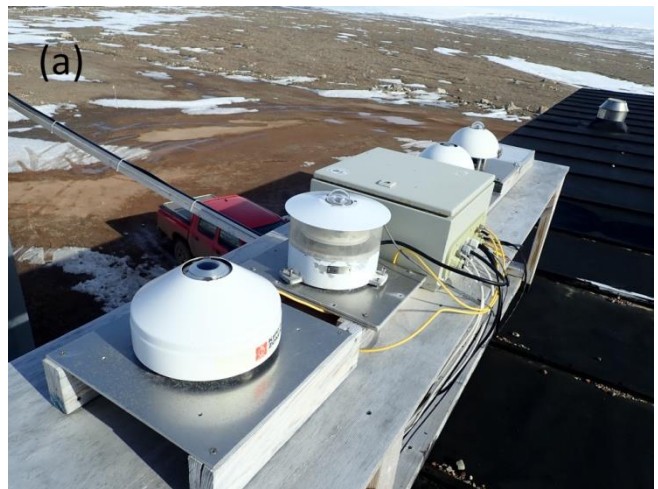 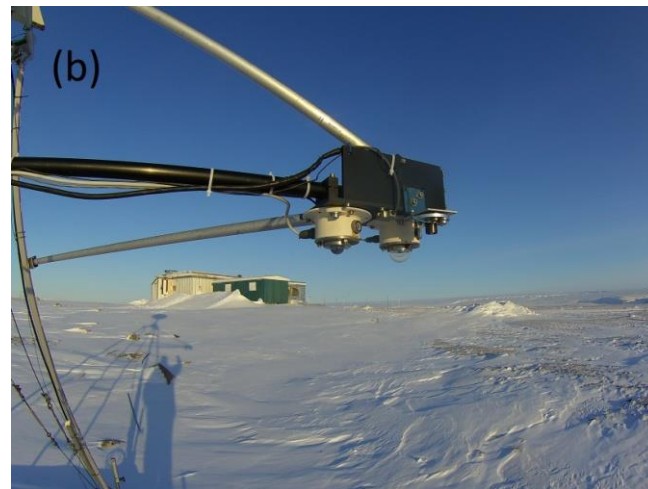

Figure 3: Pictures of (a) the upward-looking radiometers for DSI and DLI installed on the roof the building #1971 and (b) the downward-looking radiometers measuring USI and ULI.

Once a week local technicians or military personnel check on the instruments installed at the THAAO and clean the radiometers' domes. Twice a year, research personnel from the involved institutions travel to the observatory to perform instrument maintenance (e.g., radiometers intercalibration) and intensive measurement campaigns.

Figure 2 summarizes the model and the serial number of the radiometers, and the corresponding measured parameters. During some periods, as in 2013-2015, 2018 and 2019, pairs of radiometers have been simultaneously operated to assess the

185 behaviour of the different instruments and to verify their calibration. Overlaps for shorter intercalibrations also occurred in other periods (see Sections 2.1.3 and 2.1.4).

In this paper, we discuss measurements made since 2009, for which we have a good record of instrument characterization and calibration traceability.

Few interruptions in the time series of DSI and DLI have occurred since 2009. No DSI measurements have been collected

from September 2009 to February 2011. Other data gaps due to instrumental failures are present in the periods from December 2017 to mid-January 2018, from the end of August to mid-October 2018, from May to mid-July 2019, and from mid-May to mid-June 2020.

Surface radiation measurements are complemented by continuous observations of several atmospheric parameters: meteorological variables (pressure, temperature, and relative humidity) at the ground (see Section 2.1.1), integrated water

vapour, cloud liquid water path, tropospheric profiles of temperature and humidity from an RPG Humidity And Temperature PROfiler (HATPRO-G2), infrared zenith sky brightness temperature, sky images since 2016, profiles of aerosol and clouds from a Lufft 15k ceilometers since 2020, aerosol optical depth and aerosol properties from an AERONET Cimel since 2007 (Holben et al., 1998; Di Biagio et al., 2012; Calì Quaglia et al., 2022), in situ PM10 sampler for chemical analyses since

2010 (Becagli et al., 2020). A water Vapor Emission Spectrometer for Polar Atmosphere (VESPA-22) measures
stratospheric and mesospheric vertical profiles of water vapour and the respective column integrated value since 2016 (Mevi et al., 2017).

In addition, summertime integrated column amounts and vertical profiles of trace gases are derived with an FTIR spectrometer operated by the US National Center for Atmospheric Research since 1999 (Hannigan et al., 2009). Event-oriented lidar measurements and radiosoundings are also performed. The lidar system, installed in 1990, was initially developed for the detection of changes in the upper atmospheric temperature profiles linked with the evolution of the polar vortex and the formation of polar stratospheric clouds (Di Biagio et al., 2010; di Sarra et al., 2002; di Sarra et al., 1992). In 2009, the system was updated with additional channels dedicated to the backscattered signals from tropospheric aerosols.

More details on the instruments installed at THAAO are available on the dedicated website (https://www.thuleatmos-it.it).

### 2.1.1 Meteorological station

A dedicated meteorological station was installed in July 2016 on one side of the building, at an altitude of about 4 meters a.g.l.

A Campbell Scientific HC2S3 probe covered by a radiation shield measures temperature (with a PT100 RTD sensor) and relative humidity (with a ROTRONIC Hygromer IN1 sensor). The accuracy provided by the manufacturer at 23 °C is ±0.1 °C for temperature measurements and ±0.8% for relative humidity. The atmospheric pressure is measured by a Campbell Scientific CS100 barometer with accuracy of ±1.0 hPa for temperatures from 0° to 40°C, and ±1.5 hPa from -20° to +50°C. Data are collected by a Campbell datalogger CR200X every 10 minutes until the end of January 2022, every minute afterwards.

### 2.1.2 Pyranometers and pyrgeometers: main characteristics

According to the World Meteorological Organization definition (WMO, 2021), Eppley PSP pyranometers are good quality radiometers, while Kipp&Zonen CMP21 are high-quality pyranometers. Table 1 summarizes some of the characteristics of the two types of instruments, but it is worth noticing that specifications are generic for PSPs and are not provided for each instrument, while the temperature dependence and the cosine response are determined at the factory for each CMP21 instrument.

Table 1. Characteristics of Eppley PSP and Kipp&Zonen CMP21

|  | **Eppley PSP** | **Kipp&Zonen CMP21** |
|---|---|---|
| **Level** | Good quality | High quality |

| Spectral interval | 0.285 - 2.8 µm | 0.310 - 2.8 µm |
|---|---|---|
| Temperature response | ±1% (-20 / +40 °C) | < 1% (-20 / +50 °C) |
| Directional response (normal incident beam of 1000 Wm$^{-2}$ irradiance) | ±1% (0 - 70°) and ±3% (70 - 80°) | < 10 Wm$^{-2}$ (up to 80°) |

Eppley PIR pyrgeometers are sensitive to longwave radiation between approximately 3.5 and 50 µm and are provided with
two thermistors (typeYSI 44031) to measure the temperature of the case and of the dome in order to correct for the window heating effect (Philipona et al., 1995). The nominal temperature dependence varies in the range ±1% between -20 and +40°C, but this feature is generic to all PIRs and is not determined for each instrument.

Kipp&Zonen CGR4 pyrgeometers measure LW radiation in a slightly different range, between 4.5 and 42 µm. The window heating effect is minimized by a special meniscus shape dome with optimal thermal contact with the instrument case, and
only the case temperature is measured (Marty et al., 2003). The temperature dependence of the sensitivity, generally tested between -20 °C and +40 °C, is determined at the factory for each radiometer and is nominally <1%. The temperature dependence of the CGR4 sn 120550 installed at the THAAO has been measured down to -40 °C at the factory to match the extreme temperatures characterizing the Arctic winters.

The WMO (2021) and the factories' manuals provide a complete description of the pyranometers and pyrgeometers
characteristics.

All irradiance and instruments' temperature measurements are acquired by a Campbell datalogger. For DSI and DLI measurements model CR10X was adopted until January 2018 and model CR1000 afterward, whereas model CR6 is used for USI and ULI. Downward irradiances have been measured every minute until February 2018, and every 30 seconds afterwards; upward irradiances are measured every 30 seconds.

In the first step of data processing, DSI and USI are obtained dividing the datalogger voltage signal by the pyranometer sensitivity, while DLI and ULI are calculated following Eq. (1) and (2) in Section 2.1.5, used for PIR and CGR4, respectively. The instantaneous measurements by the Kipp&Zonen CMP21 and CGR4 radiometers are corrected for the temperature dependence of the sensitivity, which is provided in the calibration certificate at fixed instrument temperatures. These values are fitted with a sixth order polynomial in order to apply the correction for all measured case temperatures.

In order to improve accuracy, pyrgeometers should be mounted on a solar tracker equipped with a shading ball therefore preventing direct solar radiation from producing differential heating of the instrument dome (Marty et al., 2003). At THAAO, however, pyrgeometers are unshaded to reduce complications in the measurements. The effect of solar irradiance on PIR measurements has been analyzed using data collected at the ENEA Climate Observatory located on the island of Lampedusa, in the Central Mediterranean (https://www.lampedusa.enea.it), by Meloni et al. (2012). They found that under

cloud-free conditions with mid-latitude high levels of global solar irradiance reaching peak values of 1050 Wm$^{-2}$ in summer (5-minute average), the PIR overestimation in DLI may be as large as 10 Wm$^{-2}$. However, maximum solar irradiance values at THAAO during cloud-free conditions are around 650 Wm$^{-2}$, similar to those measured at Lampedusa during winter, implying a possible maximum overestimation of DLI by about 5 Wm$^{-2}$ in summer.

### 2.1.3 Thermal offset correction of SW irradiance measurements

After the correction for the temperature dependency of the sensitivity (CMP21 only), the thermal offset (TO) of the PSP and CMP21 upward-looking pyranometers is corrected using the thermopile signal of the co-located pyrgeometer according to Dutton et al. (2001). This method requires that simultaneous pyranometer/pyrgeometer measurements are carried out during nighttime to infer the correction to be applied during daytime. This poses some limitations at high latitudes, when alternation of daytime/nighttime periods within 24 hours occurs during limited portions of the year.

Thus, TO is calculated in different periods of the year: from January to February and from November to December, when the sun is almost always below the horizon, and from March to April and from September to October, i.e., in the periods when the sun falls below the horizon, specifically for solar zenith angle (SZA) above 95°. The TO is not calculated from May to August, when the sun is nearly always above 95° SZA.

The TO correction (unitless) is calculated as the ratio of the average SW irradiance and the average pyrgeometer's
thermopile signal (V/C in Equation 1) during nighttime, and then multiplied by instantaneous pyrgeometer's thermopile signal to obtain the TO in Wm$^{-2}$.

Table 2 shows the derived nighttime TO averages per period for all the pyranometers deployed at THAAO. The TO values are larger for PSP with respect to CMP21. It is worth noting that TO depends not only on the intrinsic thermal capacity and
structure of the instrument body but also on the temperature difference between the instrument and the environment (expressed by the pyrgeometer net radiation used to correct the TO). Thus, seasonal and interannual differences may depend on both instrumental and atmospheric characteristics (Philipona, 2002). A slight increase in the absolute values of the mean TO is detected in September-October in the THAAO dataset for the PSP pyranometers.

Table 2. Nighttime average thermal offset (TO) values (in Wm$^{-2}$) of the upward-looking pyranometers calculated in the periods of nearly absence of SW radiation (i.e., January-February and November-December) and when the sun is below the horizon (i.e., March-April and September-October)

| | 2009 | 2010 | 2011 | 2012 | 2013 | 2014 | 2015 | 2016 | 2017 | 2018 | | 2019 | | 2020 | 2021 | 2022 |
|---|---|---|---|---|---|---|---|---|---|---|---|---|---|---|---|---|
| **Instrument model and serial number** | PSP sn 33600F3 | | PSP sn 34891F3 | PSP sn 33504F3 | PSP sn 33504F3 | PSP sn 34891F3 | PSP sn 34891F3 | PSP sn 34891F3 | PSP sn 34891F3 | PSP sn 34891 F3 | CMP21 sn 160631 | PSP sn 34891 F3 | CMP21 sn 160631 | CMP21 sn 160631 | CMP21 sn 160631 | CMP21 sn 160631 |

| TO January-February | -1.3±1.0 | | -1.5±0.5 | -1.6±0.6 | n.a. | -3.5±0.9 | -3.4±1.1 | -3.6±1.0 | -3.6±1.1 | -3.7±1.2 | n.a. | -3.8±1.1 | -0.6±1.1 | -0.5±0.7 | -0.5±1.0 | -0.5±0.8 |
|---|---|---|---|---|---|---|---|---|---|---|---|---|---|---|---|---|
| TO March-April | -1.8±1.1 | n.a. | -1.7±0.6 | -1.4±0.4 | -2.0±0.7 | -3.6±1.0 | -3.6±1.0 | -3.6±1.1 | -3.3±1.0 | -3.2±1.0 | -0.3±0.7 | -3.8±1.0 | -0.5±0.7 | -0.5±0.8 | -0.5±0.6 | -0.5±0.7 |
| TO September-October | n.a. | n.a. | -1.8±0.7 | -2.0±0.8 | -2.1±0.7 | -4.0±1.1 | -4.3±1.4 | -4.3±1.1 | -4.0±1.2 | -4.2±1.0 | -0.5±0.6 | -6.2±1.7 | -0.7±0.8 | -0.5±0.7 | -0.5±0.8 | -0.7±1.3 |
| TO November-December | n.a. | | -1.6±0.7 | -1.8±0.7 | -2.0±0.6 | -3.9±1.1 | -3.6±1.1 | -3.9±1.2 | -3.9±1.4 | -3.9±1.2 | -0.5±0.8 | -7.3±1.6 | -0.5±0.9 | -0.4±0.9 | -0.5±0.8 | -0.7±0.8 |

The TO correction obtained as the mean of the values in March-April and September-October is applied to measurements from May to August.

The TO of the downward-looking PSP sn 33599F3 has also been calculated. Since the instrument faces a surface whose temperature is not sensibly lower than its own, the TO values are negligible (Table 3), and no correction is applied.

Table 3. Nighttime average thermal offset (TO) values (in $Wm^{-2}$) of the downward-looking pyranometer calculated in the periods of nearly absence of SW radiation (i.e., January-February and November-December) and when the sun is below the horizon (i.e., March-April and September-October)

| | 2016 | 2017 | 2018 | 2019 | 2020 | 2021 | 2022 |
|---|---|---|---|---|---|---|---|
| **Instrument model and serial number** | PSP sn 33599F3 | PSP sn 33599F3 | PSP sn 33599F3 | PSP sn 33599F3 | PSP sn 33599F3 | PSP sn 33599F3 | PSP sn 33599F3 |
| **TO January-February** | n.a. | -0.1±0.6 | 0.0±0.5 | 0.0±0.5 | -0.1±0.5 | 0.0±0.5 | 0.0±0.5 |
| **TO March-April** | n.a. | 0.0±0.4 | -0.1±0.5 | 0.0±0.4 | -0.2±0.5 | 0.0±0.4 | -0.1±0.5 |
| **TO September-October** | 0.0±0.5 | 0.1±0.5 | 0.2±0.4 | 0.1±0.5 | 0.1±0.5 | 0.1±0.4 | 0.2±0.4 |
| **TO November-December** | 0.0±0.4 | 0.0±0.5 | 0.0±0.4 | -0.1±0.4 | 0.0±0.4 | 0.0±0.5 | 0.1±0.4 |

### 2.1.4 Pyranometers calibration and cosine correction

The calibration of each radiometer has been checked at the ENEA Climate Observatory in Lampedusa before its installation at THAAO. A set of calibrated instruments, including Kipp&Zonen pyranometers CMP21 and CMP22 models, and a YES Inc. TSP-700, is maintained at Lampedusa. All radiometers are ventilated and routinely cleaned to ensure the removal of dew and dirt.

As recommended by the factory, each reference pyranometer is calibrated at least every two years at the Physikalisch-Meteorologisches Observatorium Davos/World Radiation Center (PMOD/WRC), where they are compared to reference instruments traceable to the World Radiometric Reference as recommended by the WMO. The response of the other pyranometers is then checked by comparison with the newly calibrated instrument by co-location on the roof of the Lampedusa Observatory.

The irradiances of PSP serial numbers 33504F3, 33599F3, 33600F3, and 34891F3, have been compared with those of a calibrated CMP21 at Lampedusa before installation at THAAO. This comparison, carried out during cloud-free days, also allowed the estimation of the cosine response of each pyranometer, by calculating the ratio of the two irradiances as a function of the SZA. This feature is critical in polar regions, where high SZAs require an adequate characterization of the pyranometer response as a function of the solar elevation.

Figure 4 shows the comparison of the DSI from CMP21 sn 090206 and PSP sn 34891F3 during some cloud-free days in spring-summer 2012 in Lampedusa. The cosine response of the CMP21, in its turn, is estimated by the factory to be very good (deviations not larger than 0.7% up to 60° zenith angle and 1.2% up to 80°) and has been assessed against two other pyranometer models, Kipp&Zonen CMP22 and YES Inc. TSP-700: both models have a superior cosine response compared to PSP and CMP21.

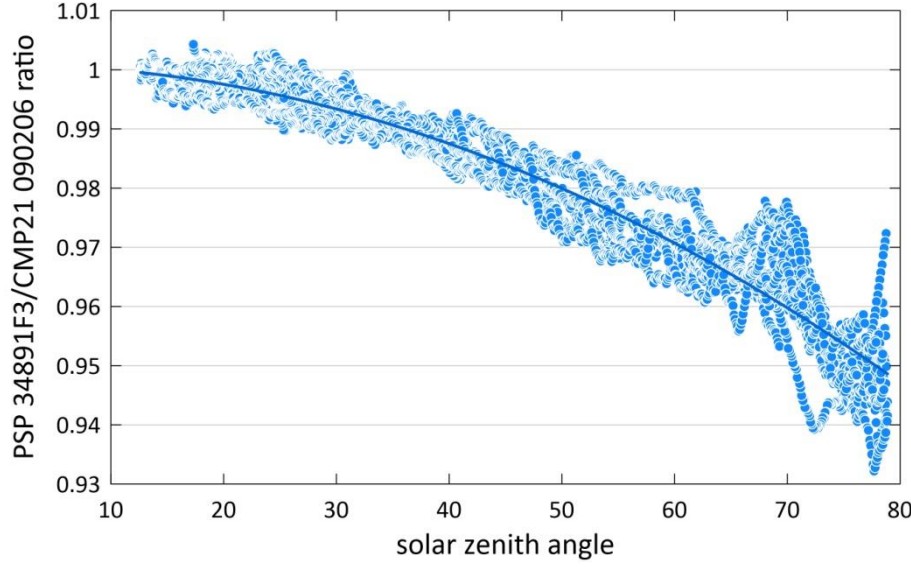

Figure 4: Ratio of the DSI measured by the PSP sn 34891F3 and by the CMP21 sn 090206 as a function of the solar zenith angle.

After correcting for the different sensitivities of the two instruments and TO, the change in the ratio with the SZA is computed. Figure 4 shows that the cosine response of the PSP can be very different from the generic factory specifications, thus an ad hoc characterization of the instrumental performances is necessary to improve measurement accuracy.

A second-order polynomial fitting curve is used to correct PSP measurements according to the derived cosine response. The USI measurements of the PSP sn 33599F3 successive to the installation on the mast are not cosine corrected.

After the installation at THAAO, the temporal stability of the radiometers' calibration has been assessed by on-site comparison with newly calibrated radiometers. PSP sn 34891F3 was checked in June 2016 against the newly calibrated PSP sn 33599F3 to be installed for USI measurements.

In March 2018 the new CMP21 sn 160631 was co-located with the PSP sn 34891F3 at THAAO and the outputs of the two instruments were compared for several days. The PSP's sensitivity was updated to compensate for a 6% reduction compared

to 2012. To account for this behaviour, a linear variation with time was then applied to the PSP sensitivity starting from 2012.

The sensitivity of the CMP21 sn 160631 and of the PSP sn 33599F3 has been assessed in August 2021 and in April 2023 against CMP21 sn 170832 calibrated at PMOD/WRC, showing no significant changes.

**2.1.5 Pyrgeometers calibration**

Pyrgeometers also are calibrated at PMOD/WRC by comparison to the reference, the World Infrared Standard Group (WISG) of four pyrgeometers, during nighttime under cloudy and cloud-free conditions.

PIR sn 33499F3 was calibrated in 2006 at the Lampedusa Observatory using as reference PIR sn 33500F3, modified with three thermistors inside the dome and calibrated at PMOD/WRC. Meloni et al. (2012) developed a methodology to transfer

the calibration from a newly calibrated pyrgeometer using the most accurate formulas for the calculation of DLI, as Albrecht and Cox (1977) and Philipona et al. (1995), using only nighttime measurements. The formula used to compute the DLI from the PIR signals is from Albrecht and Cox (1977):

$$DLI = \frac{V}{C} + k_2\sigma T_{case}^4 - k_3\sigma(T_{dome}^4 - T_{case}^4) \qquad (1)$$

where V is the thermopile signal (in mV), Tcase and Tdome are the instrument body and dome temperature (in K) computed

from the measurements of the respective thermistors using the Steinhart and Hart equation, C is the sensitivity (in mV W$^{-1}$ m$^2$), $k_2$ and $k_3$ are coefficients taking into account corrections for body and dome thermal emissions, and $\sigma$ is the Stefan-Boltzmann constant. C, $k_2$ and $k_3$ are determined during intercalibrations.

The calibration of PIR sn 33499F3 was tested in 2008 at Lampedusa before being moved to THAAO, and in 2010 at THAAO by comparison with the new CGR4 sn 090107 that was installed at the observatory next to the PIR for a few days in October. Before being installed to carry out ULI measurements, the PIR was checked on site in June 2016 and in August 2021: no significant changes in the sensitivity were detected.

CGR4 sn 120550 was factory calibrated in 2012 and then installed at THAAO next to PIR sn 33499F3 in February 2013, and it served also as a new calibration for the PIR. Successive on-site CGR4 checks were performed in June 2016, November 2019, August 2021, and April 2023, showing no changes in the sensitivity.

The formula for the computation of DLI from the CGR4 measurements with the factory calibration constant is:

$$DLI = \frac{V}{C} + \sigma T_{case}^4 \tag{2}$$

As an example, Figure 5a shows the scatterplot of PIR and CGR4 irradiances during the 2013 intercalibration campaign before and after applying the new coefficients. The differences of DLI are also plotted (Figure 5b). The mean bias and standard deviations decrease from $3.0\pm1.9$ Wm$^{-2}$ to $0.0\pm1.0$ Wm$^{-2}$ when the new coefficients are used. It is worth noting that the largest differences occur for the lowest DLI values, typical of the polar environment and under cloud-free conditions. This behaviour is expected to be attributed to the different dependence of the sensitivity with respect to the body temperature of the two instruments.

Since it is not possible to propagate in time the evolution of the different components of the DLI irradiance formulas, the coefficients found during the 2013 intercomparison were adopted for the whole installation period at THAAO (2009-2013).

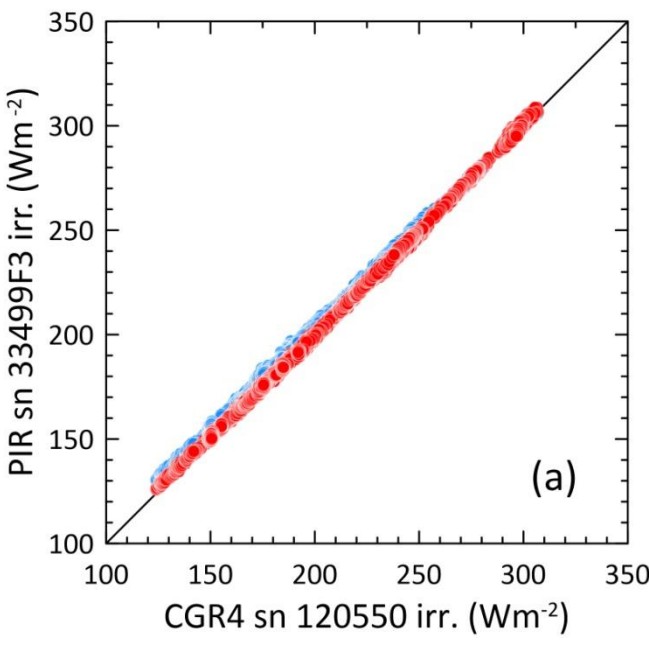
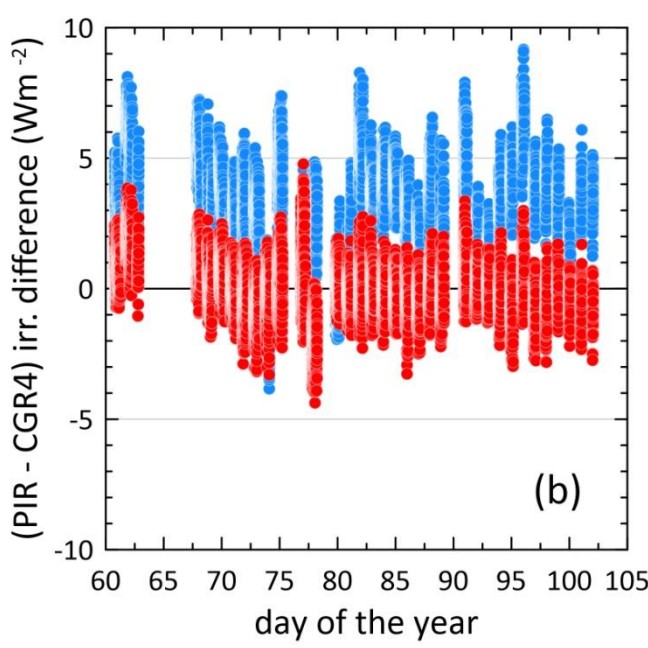

Figure 5: (a) Scatterplot of the DLI data collected simultaneously by the PIR sn 33499F3 and the CGR4 sn 120550 in 2013 and (b) temporal evolution of the DLI difference. Blue circles are the data obtained using the original PIR factory calibration in 2006, while red circles are obtained with the calibration factor calculated after the intercomparison.


### 2.1.6 Uncertainties and quality checks

The WMO (2021) defines the expected maximum uncertainty on hourly data, excluding calibration errors, from different types of pyranometer: good quality instruments have an 8% uncertainty, while high quality ones have a 3%. According to the Kipp&Zonen calibration certificate, the expanded uncertainty (two standard deviations) resulting from the calibration of the

CMP21 sn 160631 pyranometer is ±1.41%. For the calibration of the PSP whose cosine response has been empirically determined, the estimated expanded uncertainty is about ±2%. According to Kipp&Zonen calibration certificate, the expanded uncertainty on the sensitivity of the calibrated CGR4 sn 120550 pyrgeometer is about ±3.4%. The measurement uncertainties of a standard pyrgeometer calibrated at the PMOD/WRC is ±2.3 Wm$^{-2}$ (Gröbner et al., 2009). When including the uncertainty due to the acquisition system, the overall expanded uncertainty can be assumed ±5 Wm$^{-2}$ on LW irradiance

measurements (Meloni et al. 2015). However, uncertainties are larger for unshaded instruments (see Section 2.1.2).

Measurements are quality checked following the recommendations adopted for BSRN stations (Long and Dutton, 2002; Long and Shi, 2008). In particular, tests for the "physically possible limits" are performed on DSI and USI data at their native time resolution: such tests fix at -4 Wm$^{-2}$ the lower limit for DSI and USI measurements. However, Long and Shi (2008) emphasize that -4 Wm$^{-2}$ for nighttime values may suggest that the thermal offset has not been properly corrected; for

this reason, we adopted a minimum value for DSI and USI of -2 Wm$^{-2}$, that is the threshold for "extremely rare limits" discussed by Long and Shi (2008). Tests on DLI and ULI are performed applying these "extremely rare limits".

The minimum and maximum limits applied to the tests are reported in Table 4.

Table 4. Limits applied to the quality check tests. $S_a$ is the solar constant at mean Earth-Sun distance (assumed as 1368

Wm$^{-2}$) adjusted to the effective Earth-Sun distance.

|  | DSI (Wm$^{-2}$) | USI (Wm$^{-2}$) | DLI (Wm$^{-2}$) | ULI (Wm$^{-2}$) |
|---|---|---|---|---|
| **Min** | -2 | -2 | 60 | 60 |
| **Max** | $S_a$ x 1.5 x cos(SZA)$^{1.2}$ + 100 | $S_a$ x 1.2 x cos(SZA)$^{1.2}$ + 50 | 500 | 700 |

The 100% of the DLI and ULI values are within the "extremely rare limits". DSI shows a small percentage of values below the minimum limit: the percentage of data outside the -2 Wm$^{-2}$ threshold for each year varies between 0.5% and 4.4% for the

PSPs and between 0.3% and 1% for CMP21. A much lower occurrence of data below the -2 Wm$^{-2}$ threshold for USI is found, between 0.1% and 0.2%. Data falling outside the test boundaries are rejected in the dataset used in this analysis.

In addition, tests are performed to compare DLI and ULI with limits defined according to air temperature and the Stefan-Boltzmann law, as described in Table 5. This quality check, routinely applied to BSRN data (Long and Dutton, 2002), is executed for the years when continuous meteorological parameters are measured, i.e., since July 2016 (see Section 2.1.1).


Table 5. Limits applied to the quality check tests for DLI and ULI as a function of air temperature. $T_a$ is the air temperature (K).

|  | DLI (Wm$^{-2}$) | ULI (Wm$^{-2}$) |
|---|---|---|
| **Min** | $0.4 \times \sigma T_a^4$ | $\sigma (T_a - 15)^4$ |
| **Max** | $\sigma T_a^4 + 25$ | $\sigma (T_a + 25)^4$ |

Five-minute means of DLI and ULI are used to carry out the tests, since air temperature is measured every 10 minutes from July 2016 to January 2022. Meteorological data are linearly interpolated to the radiation data time resolution. Starting in January 2022, five-minute means of DLI, ULI, and air temperature are calculated. Figure 6 provides an example of the test for the year 2021. 100% of the DLI and ULI data in the period 2016-2022 satisfies the test and is retained.

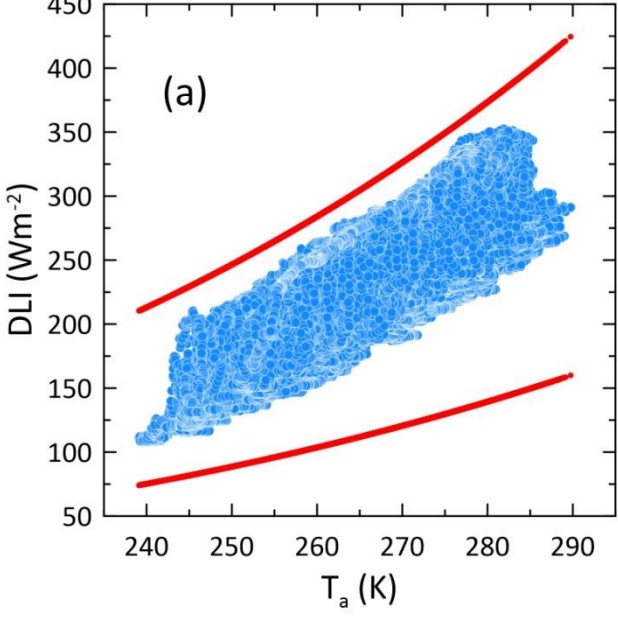
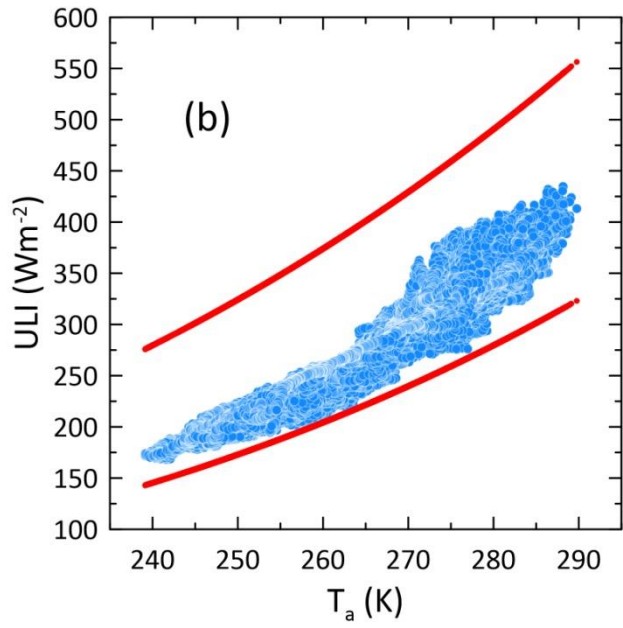


Figure 6: Graphic representation of the BSRN test for (a) DLI and (b) ULI versus air temperature for the year 2021. Blue circles represent measurements, while red circles the test limits calculated according to the formulas in Table 5.

## 3 Results

In the following analysis, hourly, daily, and monthly means are computed. Daily means are calculated from hourly means, and monthly means from the daily ones (Roesch et al., 2011). Averages obtained using too few samples are discarded. The minimum timespans are set at 45 minutes, 18 hours, and 22 days, respectively, for hourly, daily, and monthly means. Seasons are defined by grouping the months as follows: spring (MAM), summer (JJA), autumn (SON), winter (DJF). The percent of valid hourly means available per year and per irradiance component are reported in Table 6. The numbers

reflect the fact that radiometers were not installed in some periods, the unavailability of data due to datalogger interruptions, and quality rejected data.

Table 6. Percent of valid hourly means per year and irradiance component

| Year | DSI | DLI | USI | ULI |
|---|---|---|---|---|
| 2009 | 54.2 | 89.5 | | |
| 2010 | - | 99.5 | | |
| 2011 | 83.2 | 97.2 | | |
| 2012 | 65.4 | 67.3 | | |
| 2013 | 80.7 | 82.7 | | |
| 2014 | 95.0 | 99.8 | | |
| 2015 | 95.3 | 100.0 | | |
| 2016 | 93.9 | 99.8 | 46.2 | 46.2 |
| 2017 | 88.1 | 91.5 | 98.0 | 98.0 |
| 2018 | 82.0 | 91.3 | 99.8 | 100.0 |
| 2019 | 78.3 | 80.8 | 99.5 | 99.6 |
| 2020 | 92.1 | 93.0 | 100.0 | 100.0 |
| 2021 | 98.8 | 99.8 | 96.4 | 96.6 |
| 2022 | 99.5 | 99.8 | 99.8 | 99.8 |


## 3.1 Air temperature

The analysis of the temporal evolution of the air temperature is useful to understand the variability of the longwave irradiance components, so the main characteristics are presented here.

The time series of daily and monthly means of $T_a$ and $T_a$ anomaly, calculated using the 2016-2022 average, are plotted in Figure 7, while Table 7 presents the monthly and seasonal statistics (mean, standard deviation, minimum, and maximum) over the 2016-2022 period.

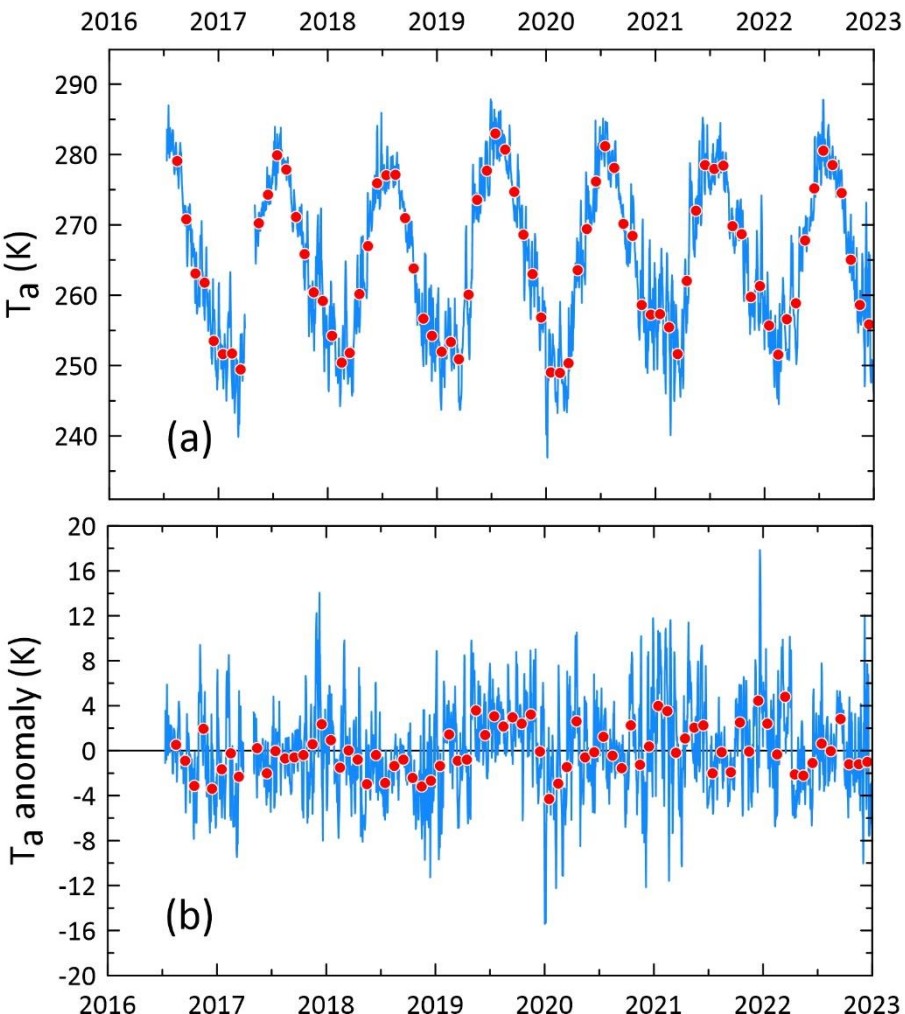

Figure 7: Time series of (a) air temperature and (b) air temperature anomaly calculated by subtracting the 2016-2022 means. The blue line represents daily mean values, while red circles mark monthly means.

Table 7. Monthly and seasonal air temperature means, standard deviation, minimum and maximum (units of K).

| | $T_a$ mean | $T_a$ st.dev. | $T_a$ min. | $T_a$ max. |
|---|---|---|---|---|
| **JAN** | 253.3 | 3.0 | 249.0 | 257.3 |
| **FEB** | 251.9 | 2.3 | 249.0 | 255.5 |
| **MAR** | 251.8 | 2.5 | 249.5 | 256.6 |
| **APR** | 260.9 | 1.8 | 258.8 | 263.5 |
| **MAY** | 270.0 | 2.5 | 267.0 | 273.6 |
| **JUN** | 276.3 | 1.6 | 274.3 | 278.5 |
| **JUL** | 279.9 | 2.2 | 277.1 | 283.0 |
| **AUG** | 278.5 | 1.1 | 277.2 | 280.7 |
| **SEP** | 271.7 | 2.0 | 269.8 | 274.7 |
| **OCT** | 266.2 | 2.4 | 263.1 | 268.7 |
| **NOV** | 259.8 | 2.1 | 256.6 | 263.0 |
| **DEC** | 256.9 | 2.7 | 253.5 | 261.3 |
| | | | | |
| **MAM** | 261.0 | 0.9 | 259.7 | 261.9 |
| **JJA** | 278.2 | 1.3 | 276.7 | 280.4 |
| **SON** | 265.9 | 1.6 | 263.8 | 268.8 |
| **DJF** | 254.1 | 2.1 | 251.6 | 256.7 |


The monthly mean temperatures exhibit an annual cycle with a maximum in July and a minimum in February-March. Monthly values are above the freezing temperature during the summer months. Overall mean values have been calculated over the 2016-2022 period; deviations of single monthly or daily averages from the overall mean are defined here as temperature anomalies.

The lowest monthly means were measured in January and February 2020 (249.0 K); consequently, winter 2019-2020 is the coldest in the record (251.6 K). The minimum daily average of 236.9 K is on 5 January 2020, corresponding to a temperature anomaly of -15.3 K. Winter 2019-2020 was characterized by an exceptionally strong and cold stratospheric polar vortex in the Northern Hemisphere, leading to the greatest ozone loss ever recorded over the Arctic (Wohltmann et al., 2020). Another consequence was an extremely positive tropospheric Arctic Oscillation, explaining a large fraction of the observed warmth

that occurred in the Southeastern United States, Europe, and Asia from January to March, while anomalous cold was registered in Canada, Greenland, and Alaska (Lawrence et al., 2020).

The largest $T_a$ monthly means were registered in July 2019 (283.0 K), with summer 2019 being the warmest in the record (280.4 K). The maximum daily mean of 287.9 K is found on 29 June 2019, corresponding to a +5.4 K anomaly. Summer 2019 and, in particular, July is among the warmest ever recorded by ground-based measurements in Greenland, that caused a significant Greenland Ice Sheet (GrIS) mass loss (Hanna et al., 2020).

Summer 2018 is the coldest in the 2016-2022 period (276.7 K), mainly because July is characterized by a persistent negative $T_a$ anomaly (-2.9 K). Although 2018 was the second warmest year since 1900, based on surface air temperature data over land north of 60° N, central Greenland experienced colder-than-average spring and summer, with also positive precipitation anomalies (Overland et al., 2018). While the NCEP/NCAR reanalysis air temperature at 925 mbar does not show the negative anomaly reaching Pituffik latitude (Figure 2 of the report by Overland et al., 2018), such a phenomenon may have been extended to a wider region.

The largest positive daily $T_a$ anomaly is measured during four consecutive days, from 20 to 24 December 2021, with mean $T_a$ and $T_a$ anomaly of 271.9 K and +16 K, respectively, and a peak of 274.2 K and +17.9 K on 21 December. Overall, the monthly mean $T_a$ anomaly of the month is +4.4 K. The positive temperature anomaly has been observed on a larger region, encompassing the entire Greenland, the Atlantic portion of the Arctic Ocean, and the eastern Canadian Arctic (Thoman et al., 2022). Similarly, large positive $T_a$ anomalies were registered from 27 November to 2 December 2017, with mean $T_a$ anomaly of +9.6 K and a peak of +11.6 K on 28 November.

As detailed in the following analysis, air temperature plays a role in the modulation of LW irradiance components (DLI and ULI) and in regulating ground characteristics, such as surface albedo.

## 3.2 Downward and upward shortwave and longwave irradiances

The overall monthly and seasonal means of the four components of the measured irradiances, with their standard deviation, minimum and maximum are reported in Table 8.

Table 8. Monthly and seasonal means for DSI, DLI, USI, and ULI (units of $Wm^{-2}$). Seasonal statistics are calculated over the seasons when means are available for all the three months in the season.

| | | JAN | FEB | MAR | APR | MAY | JUN | JUL | AUG | SEP | OCT | NOV | DEC | | MAM | JJA | SON | DJF |
|---|---|---|---|---|---|---|---|---|---|---|---|---|---|---|---|---|---|---|
| **Mean** | DSI | 0.2 | 4.9 | 53.7 | 153.1 | 257.6 | 277.0 | 230.6 | 136.8 | 65.3 | 10.8 | 0.3 | 0.2 | | 150.7 | 215.2 | 25.3 | 1.8 |
| | DLI | 178.0 | 172.5 | 174.4 | 200.4 | 238.9 | 272.9 | 291.2 | 288.7 | 254.1 | 236.6 | 210.0 | 189.9 | | 204.3 | 283.8 | 233.7 | 179.5 |
| | USI | 0.0 | 3.8 | 40.9 | 109.6 | 132.4 | 63.9 | 37.5 | 22.5 | 24.3 | 6.3 | 0.0 | 0.0 | | 94.3 | 42.7 | 10.2 | 1.3 |
| | ULI | 215.8 | 210.3 | 211.6 | 249.2 | 296.8 | 338.6 | 356.1 | 342.5 | 301.1 | 273.2 | 245.7 | 233.9 | | 252.6 | 345.4 | 273.4 | 219.9 |
| | | | | | | | | | | | | | | | | | | |
| **S. dev.** | DSI | 0.3 | 0.7 | 3.1 | 14.1 | 14.2 | 31.0 | 30.8 | 21.4 | 9.2 | 1.8 | 0.4 | 0.3 | | 15.4 | 22.6 | 3.0 | 0.2 |
| | DLI | 16.6 | 17.7 | 12.9 | 17.8 | 11.5 | 7.8 | 7.6 | 8.7 | 13.2 | 14.2 | 15.2 | 18.7 | | 8.8 | 5.4 | 7.9 | 12.9 |
| | USI | 0.1 | 0.7 | 4.1 | 9.4 | 43.4 | 18.2 | 5.6 | 3.6 | 13.2 | 2.5 | 0.1 | 0.1 | | 17.6 | 6.1 | 4.7 | 0.3 |

| | | | | | | | | | | | | | | | | | |
|---|---|---|---|---|---|---|---|---|---|---|---|---|---|---|---|---|---|
| | **ULI** | 12.5 | 10.0 | 8.7 | 7.4 | 14.4 | 9.6 | 10.9 | 5.8 | 10.6 | 12.1 | 9.4 | 12.3 | | 5.4 | 6.7 | 6.8 | 9.5 |
| | | | | | | | | | | | | | | | | | | |
| **Min.** | **DSI** | -0.2 | 3.5 | 46.1 | 116.7 | 233.1 | 224.5 | 188.2 | 94.1 | 45.0 | 6.9 | -0.2 | -0.2 | | 108.9 | 173.6 | 18.7 | 1.5 |
| | **DLI** | 150.3 | 146.6 | 160.2 | 163.8 | 219.1 | 259.8 | 280.0 | 269.5 | 237.1 | 206.6 | 189.4 | 164.3 | | 191.1 | 276.4 | 216.9 | 162.8 |
| | **USI** | -0.1 | 3.2 | 33.3 | 95.1 | 63.5 | 40.7 | 29.6 | 16.9 | 10.1 | 4.2 | 0.0 | -0.1 | | 68.8 | 34.9 | 5.8 | 1.0 |
| | **ULI** | 197.1 | 195.8 | 204.9 | 242.7 | 280.6 | 325.8 | 342.0 | 337.3 | 288.3 | 255.1 | 231.0 | 217.8 | | 245.6 | 338.5 | 264.6 | 207.6 |
| | | | | | | | | | | | | | | | | | | |
| **Max.** | **DSI** | 0.7 | 6.3 | 57.1 | 172.5 | 275.4 | 334.1 | 274.4 | 168.6 | 80.2 | 14.1 | 0.9 | 0.8 | | 163.6 | 246.9 | 29.6 | 2.1 |
| | **DLI** | 204.1 | 209.7 | 204.0 | 233.5 | 256.5 | 286.1 | 307.1 | 297.8 | 284.3 | 260.8 | 237.1 | 221.3 | | 221.2 | 295.2 | 246.1 | 198.0 |
| | **USI** | 0.0 | 5.1 | 44.5 | 119.4 | 177.4 | 85.6 | 45.1 | 27.1 | 42.4 | 11.2 | 0.1 | 0.1 | | 113.3 | 49.7 | 17.9 | 1.7 |
| | **ULI** | 233.0 | 226.7 | 228.6 | 260.7 | 319.4 | 352.9 | 367.1 | 352.4 | 313.2 | 286.5 | 259.2 | 252.1 | | 258.4 | 354.1 | 286.3 | 231.8 |

### 3.2.1 Shortwave radiation

Figure 8 displays the DSI and USI time series as daily and monthly means.

DSI is absent (SZA≥90°) from 29 October to 13 February, when the sun remains below the horizon. The period when the sun remains above the horizon (SZA<90°) throughout the day goes from about 26 April to 16 August. At the summer solstice, the minimum and maximum SZA at THAAO are 53° and 80°, respectively, with corresponding cloud and aerosol - free instantaneous DSI values of 675 Wm$^{-2}$ and 136 Wm$^{-2}$, measured on 21 June 2021.

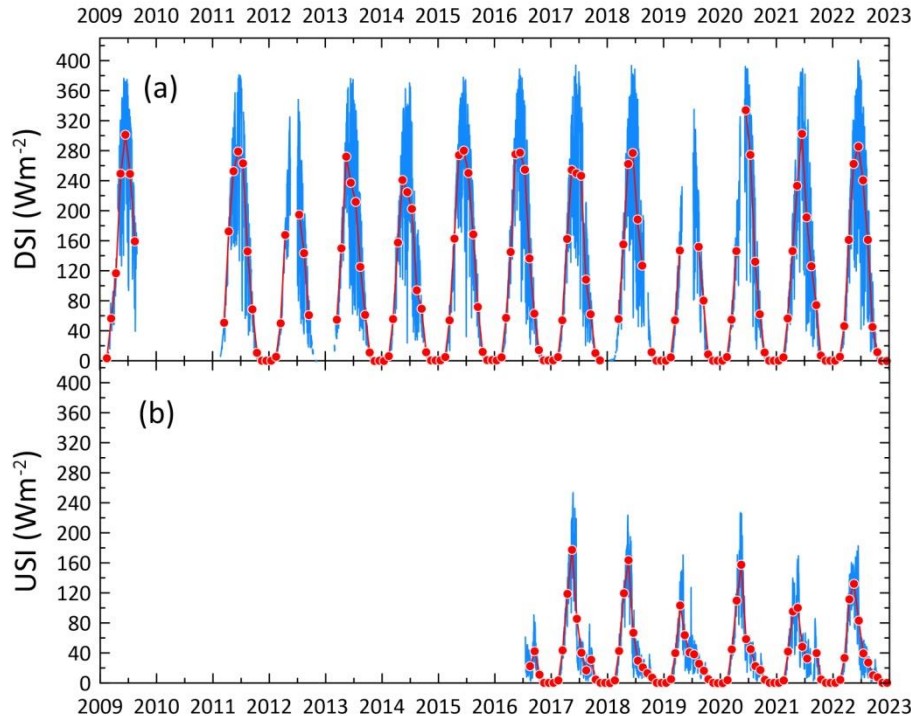

Figure 8: Time series of (a) DSI and (b) USI daily (blue line) and monthly (red circles) means.

The year-to-year variability is very marked, as it appears in the box plot of Figure 9a, as well as in the plot of the daily and monthly mean anomalies shown in Figure 10a.

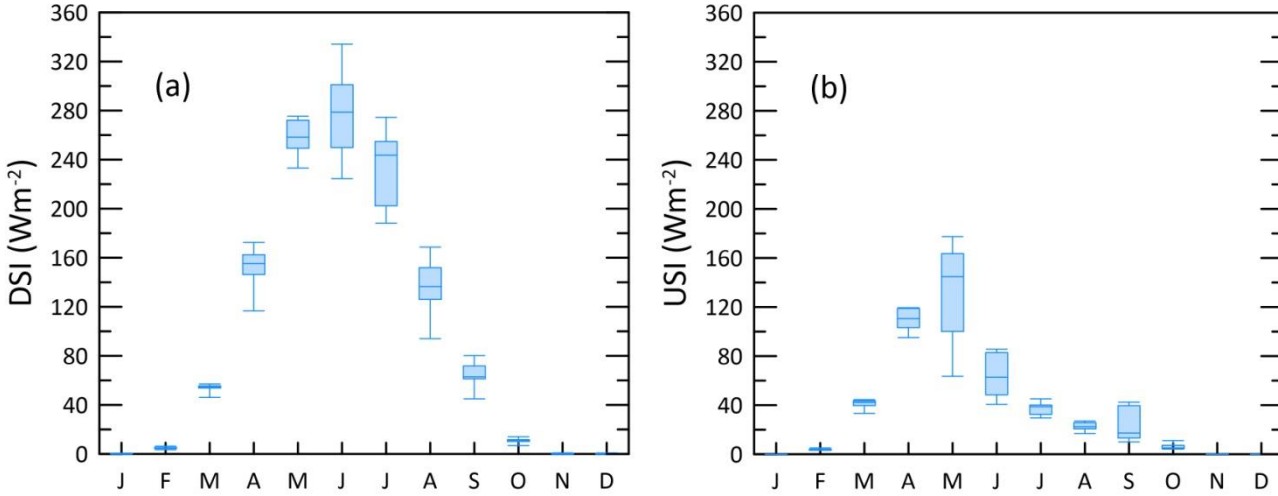

Figure 9: Box plot of the monthly means of (a) DSI and (b) USI. The median, 25% and 75% percentiles, and the minimum and maximum values are represented.

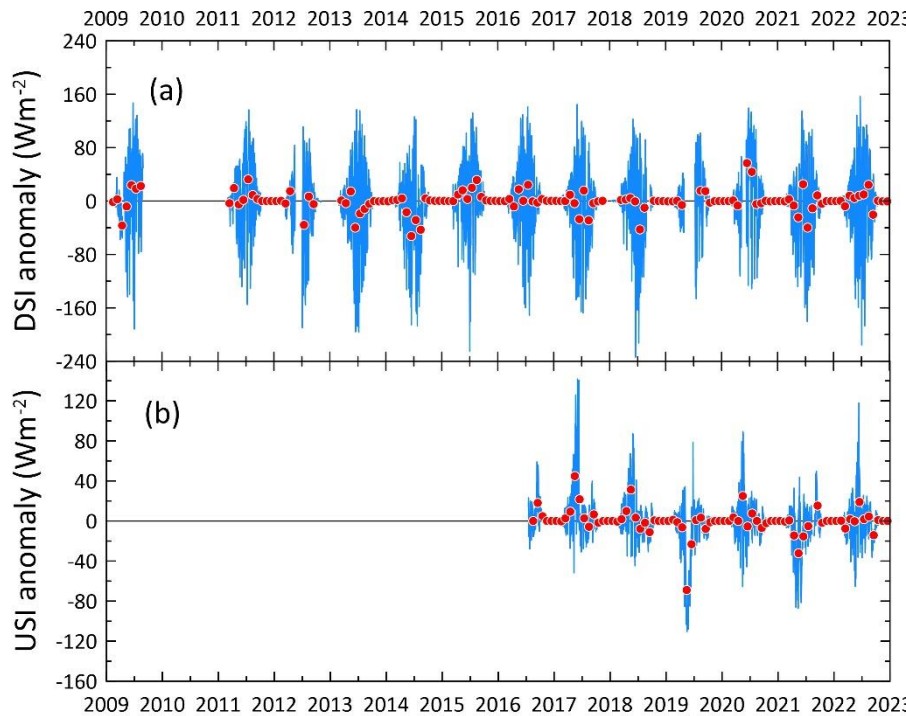

Figure 10: Time series of (a) DSI anomaly and (b) USI anomaly daily (blue line) and monthly (red circles) means.

June is the month when the largest DSI values are experienced, both as monthly mean (277.0 Wm$^{-2}$) and median (278.7 Wm$^{-2}$) values. The largest spread of the data in terms of percentiles and maximum-minimum differences is reached in June and July. This behaviour is common to summer months, when DSI levels and cloud occurrence are the highest (Shupe et al., 2011). The year 2020 is the one with the largest absolute monthly maxima for June and July, recorded at 334.1 Wm$^{-2}$ and 274.4 Wm$^{-2}$, respectively.

Within the interval 2009-2022, and disregarding the years 2010 and 2019 characterized by some missing DSI monthly means, 2020 is the year with the largest summer mean value, 246.9 Wm$^{-2}$, and 2014 the year with the lowest summer mean, 173.6 Wm$^{-2}$. The DSI anomaly mainly reflects the inter-annual difference in cloud occurrence.

Similarly to DSI, USI is absent from the end of October to mid-February, but its annual peak is anticipated in May (132.4 Wm$^{-2}$) (Figure 9b), due to the persistence of snow/ice on the ground, when DSI has already reached large values. The box height in Figure 9 indicates that large year-to-year variability occurs during this month. For example, the low May values occurred in 2019 and 2021 are linked with positive anomalies of the surface air temperatures. Figure 10b highlights the USI

anomaly in response to snow cover at the surface: for example, the strong negative anomaly of May and June 2019 reflects the anticipated melt season occurred in Wester Greenland (see Section 3.2.2).

### 3.2.2 Surface albedo

The USI/DSI ratio provides the shortwave surface albedo (A).  A depends not only on surface type, but also on its properties: for example, the albedo of snow/ice depends on its thickness, density and grain size, that in turn are affected by atmospheric conditions (Pirazzini, 2004). Moreover, the distribution of snow/ice on the surface may not be flat and vary in time, due to snow redistribution by wind and melt (Picard et al., 2020). In addition, diurnal A variations may result from geometric (e.g., solar zenith and azimuth angles), atmospheric (e.g., cloud cover), as well as instrumental factors, like the horizontal

mounting and cosine response of the pyranometers, and shadowing effects in some periods of the year (Wang and Zender, 2011). It is straightforward to assess that the uncertainty on A increases with very low irradiance conditions, corresponding to the largest A values. Moreover, the number of measurements on which daily mean A is calculated significantly changes during the year, due to different daytime duration (see Figure 11 for the minimum SZA reached in each month).

    Here A values are calculated as the ratio of 5-minute means of DSI and USI, averaged during daytime, in particular for

SZA≤85° to exclude measurements with very low sun. No discrimination is made on the averaged measurements for cloudiness or geometry: such an assumption is made because the analysis is intended for evaluation of the annual evolution of A and of the links with atmospheric conditions, whereas a detailed estimation of A as a function of the different factors influencing its variability is out of the scopes of the paper. The uncertainty on A values calculated from the propagation of the uncertainty resulting from the calibration of the pyranometers is 2.8% when DSI is measured by PSP model and 2.4%

when CMP21 model is employed.

    The annual evolution of the derived daily average SW surface albedo calculated for the period 2016-2022 is shown in Figure 10. The terrain is generally free from snow/ice during most of the summertime; between mid-June and the end of August, A ranges between 0.13 and 0.18, with an average value of 0.16 and little interannual variability. A closer look reveals that the beginning and the end of the snow-free period vary from year to year.

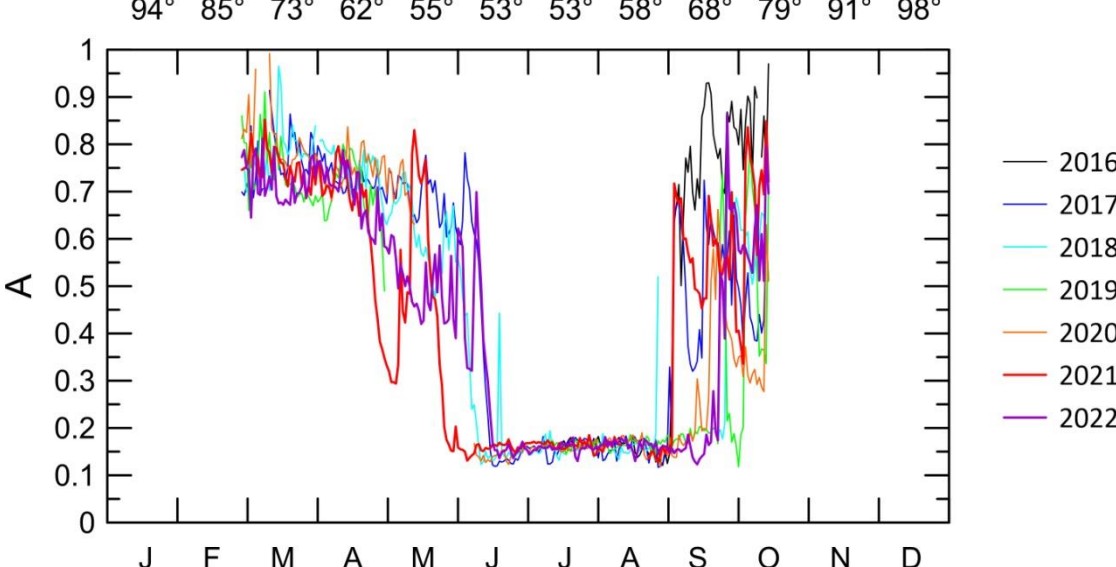

Figure 11: Annual evolution of the daily average surface albedo values from 2016 to 2022. The upper x axis indicates the minimum SZA for each month.

During months of snow-covered surface and with solar radiation, A is between 0.6 and 1.0, with limited variability from March to mid-April. In general, the conditions at the surface rapidly change at the end of May and in September, because of interannual differences in the onset/offset of the snowfall season or wind-induced transport/removal. For example, the periods from the end of April to the beginning of May and from the end of May to the beginning of June 2021 were characterized by values of A lower than average, that may be caused by reduced snowfall or liquid precipitation or surface conditions favourable to snow melt or snow removal, such as strong winds or high ground temperatures. This hypothesis is in line with the longwave fluxes measured during May 2021: DLI and ULI monthly means are the largest and the second largest, respectively, over the period 2016-2022. The onset of the snow-free surface period anticipated by several days in 2021 compared to other years. In the years 2019, 2020, and 2022, the low albedo condition continued also in September, with values <0.18.

In order to explore possible links between anticipated (in 2021) and delayed (in 2019) surface snow-free seasons and warming, the evolution of A from April to October has been related to $T_a$ and $T_a$ anomaly in the years 2019 and 2021 (Figure 12).

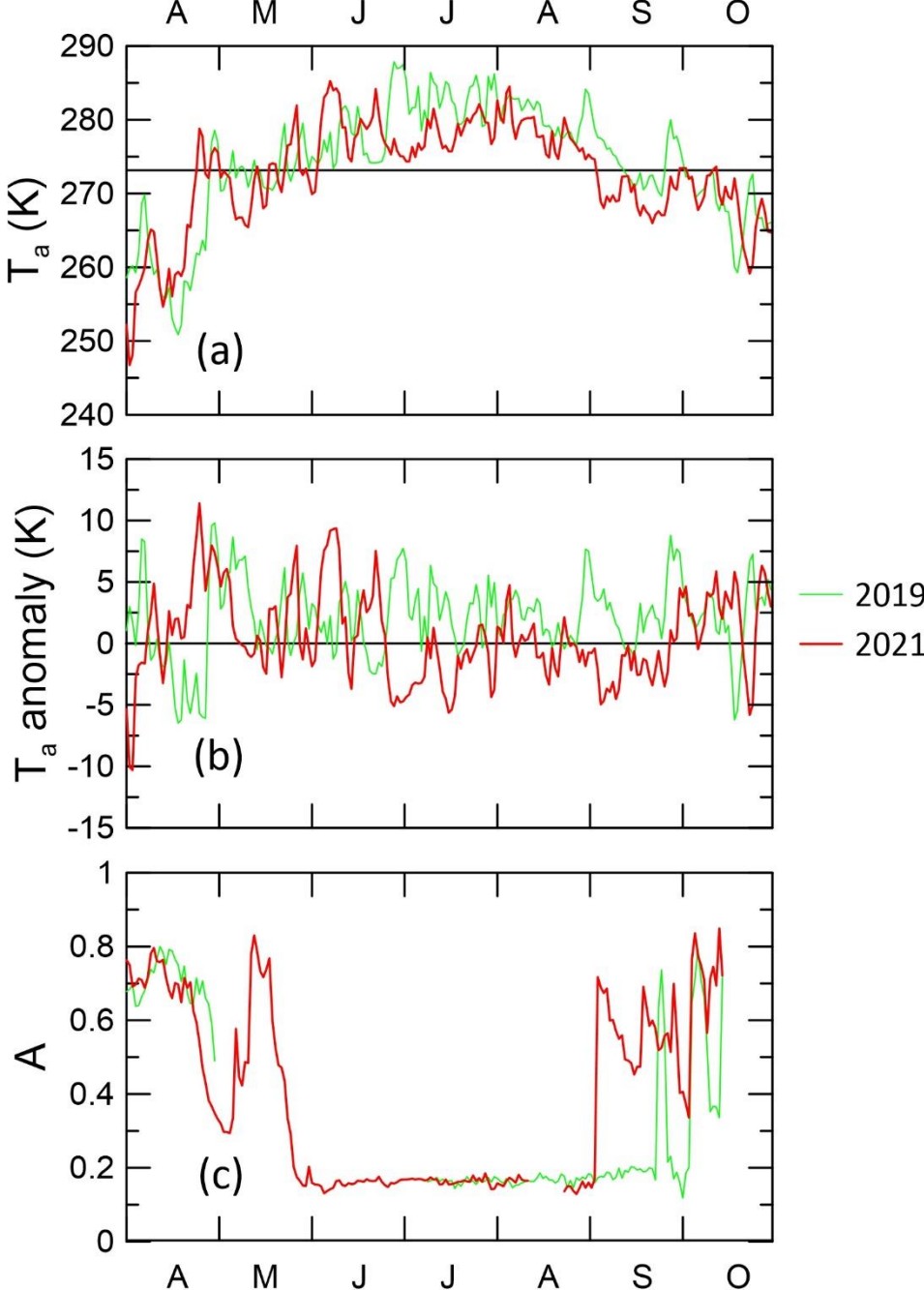

Figure 12: Evolution of daily mean (a) air temperature, (b) air temperature anomaly, and (c) surface albedo from April to October in 2019 (green line) and in 2021 (red line).

The early onset of snowmelt in mid-April 2021 is triggered by a steep increase in $T_a$ leading to values above the melting point, with a peak up to 278.8 K on 26 April, corresponding to a strong positive $T_a$ anomaly > 11 K, followed by a period of "average" conditions when A increases again reaching 0.85 before decreasing again to typical summer values.

As discussed in Section 3.1, summer 2019 is the warmest in our record, with June, July, August, and September having the largest monthly mean values measured throughout 2016-2022. A sequence of positive $T_a$ anomalies characterizes the season since the end of June; in particular, a period of positive $T_a$ anomalies with maxima of 7-8 K starts at the end of August and lasts until the end of September, when $T_a$ values remain above the melting point for six days, with a peak of 280 K on 28 September corresponding to about 9 K of $T_a$ anomaly. Even in this case air temperature is presumably responsible for the persistent snow-free conditions. Tedesco et al. (2020) documented an exceptionally melt of western Greenland in summer 2019, driven by persistent anticyclonic conditions and reduced surface albedo.

The diverse timing of the onset of the snow free period and of the snowfall and the influence on the surface albedo evolution for four Arctic observatories is highlighted by Uttal et al. (2016): they underline how snow accumulation, temperature, and cloudiness influence the timing of snowmelt, with an effect on the surface radiation budget.

### 3.2.3 Longwave radiation

DLI presents a seasonal cycle with a monthly mean maximum in July (291.2 Wm$^{-2}$) and a minimum in February (172.5 Wm$^{-2}$), when it is by far the main component of the SRB (Figure 13 and Figure 14). DLI is larger than DSI from July to April, while DSI overcomes DLI in May and June. The interannual differences in DLI (Figure 15a) are linked to the occurrence of clouds and, especially in clear sky conditions, to the air temperature and water vapor content.

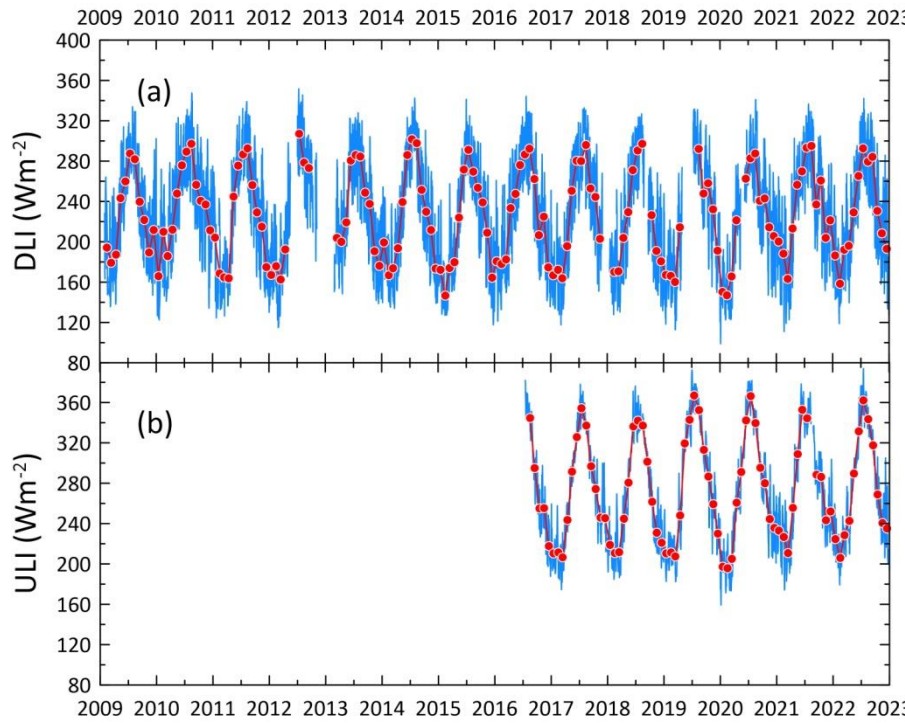

Figure 13: Time series of (a) DLI and (b) ULI daily (blue line) and monthly (red circles) means.


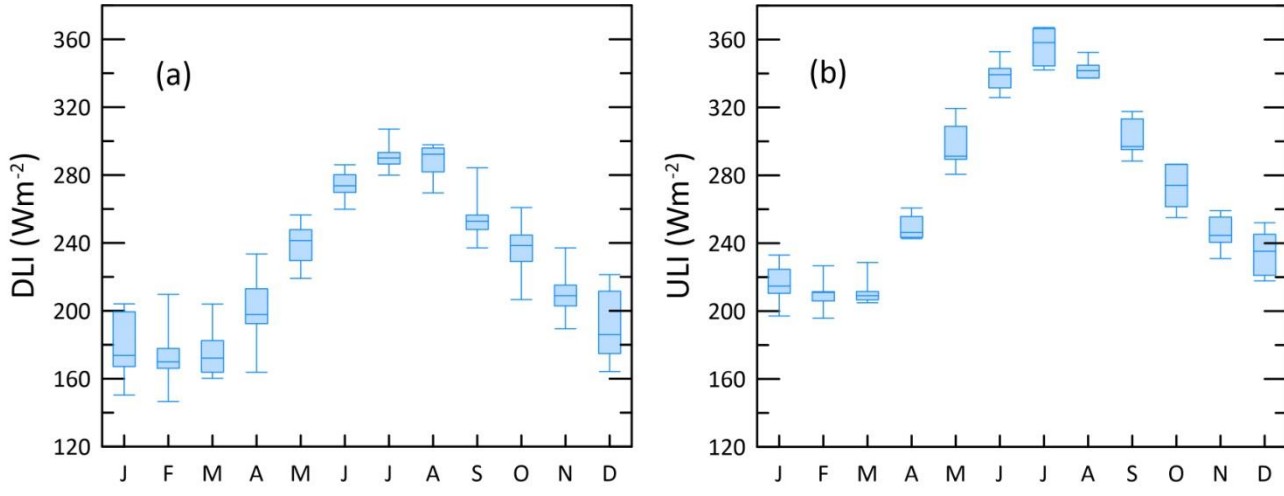

Figure 14: Box plot of the monthly means (a) DLI and (b) ULI. The median, 25% and 75% percentiles, and the minimum and maximum values are represented.

Winter 2019-2020 is characterized by the lowest DLI values (162.8 $Wm^{-2}$), mirroring the minimum $T_a$ values associated with the extraordinarily cold season in Greenland (see Section 3.1),

For example, during the days of large positive $T_a$ anomaly from 20 to 24 December 2021, large daily DLI anomalies are detected (Figure 15a), from +44 $Wm^{-2}$ (equal to +21% of the 2009-2022 mean) to +102 $Wm^{-2}$ (equal +59% of the 2009-2022 mean).

Summer 2020 stands out for its low DLI (277.6 $Wm^{-2}$) associated with very large DSI levels, implying possible prevalence of cloud-free conditions during the season, especially June and July, with their DSI peak values.

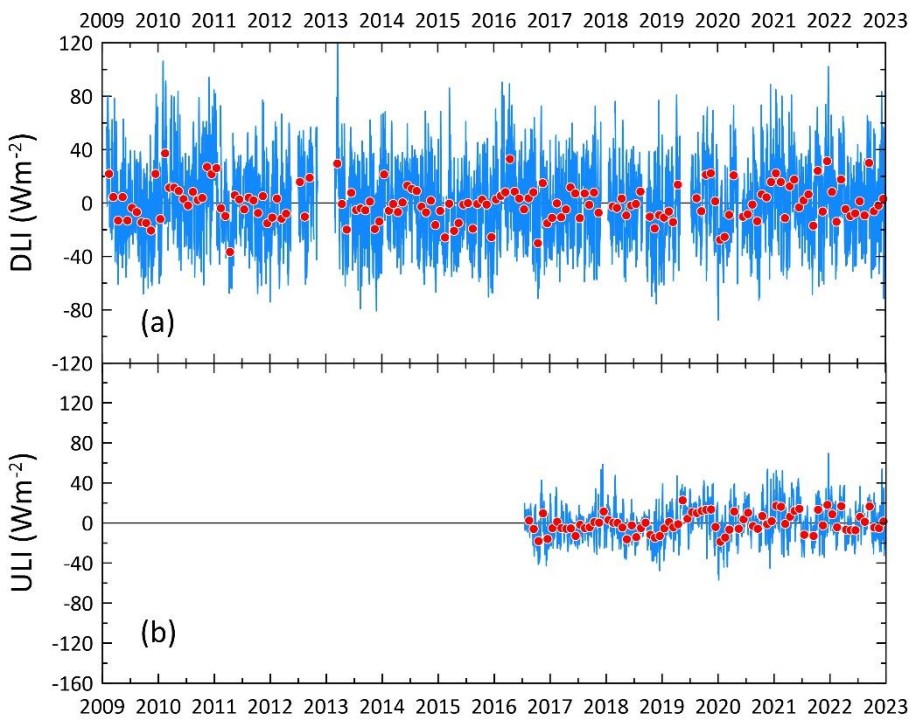

Figure 15: Time series of (a) DLI anomaly and (b) ULI anomaly daily (blue line) and monthly (red circles) means.

July 2012 is the month with the largest DLI (307.1 $Wm^{-2}$), associated with lower-than-average DSI values (194.7 $Wm^{-2}$): the two competitive effects are reasonably associated with the modulation of downwelling radiation by low-level clouds containing liquid water, that is indicated as the main cause for the enhancement of the surface melting observed over the

GrIS in July 2012 (Bennartz et al., 2013).

Examining the daily data (Figure 16), 9 out of 31 days of the month had DSI values below one standard deviation of the 2009-2022 mean, with differences reaching about 200 $Wm^{-2}$ on 4 and 5 July. Days with DLI above one standard deviation of the 2009-2022 mean are 15, with the largest difference being around 50 $Wm^{-2}$ on 4 and 5 July.

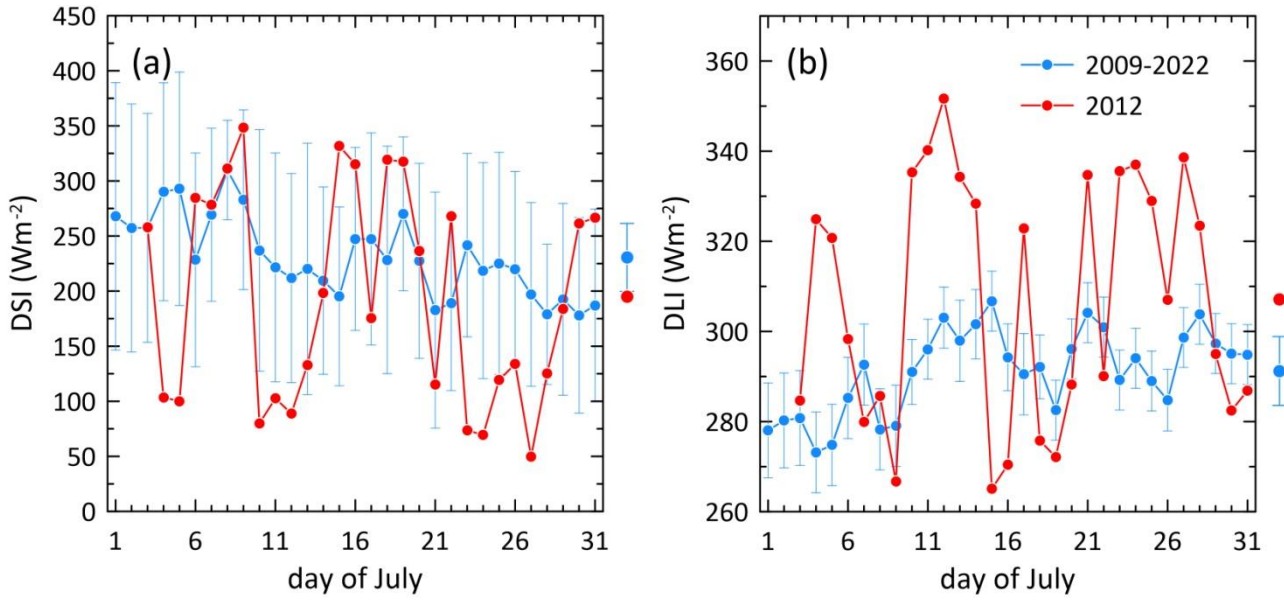


Figure 16: Daily mean (a) DSI and (b) DLI for July. Blue dots represent the 2009-2022 mean with one standard deviation error bars, while red dots represent the 2012 values. The dots outside the graphs represents the 2009-2022 monthly means with standard deviation (blue) and the 2012 monthly mean (red).


The ULI component is always larger than DLI (Figure 15b) and has a wider annual cycle, with a maximum in July (356.1 $Wm^{-2}$) and a minimum in February (210.3 $Wm^{-2}$). Moreover, ULI is generally less variable than DLI on a day-to-day basis: the reason is the dependence of ULI on ground temperature and water content, and lower sensitivity than DLI to clouds and 615 air temperature, that change more rapidly than surface properties.

As ULI is linked to surface temperature and water content, ULI anomalies reflect $T_a$ anomalies. For example, the periods of large positive $T_a$ anomalies in November-December 2017 and in December 2021 correspond to large ULI anomalies of +33.7 $Wm^{-2}$ (equal to +14% of the 2016-2022 mean) and of +53.8 $Wm^{-2}$ (equal to +23% of the 2016-2022 mean), respectively.

The measured values of ULI versus $T_a$ have been fitted with a fourth-degree polynomial curve with respect to $T_a$. A large correlation is found between the polynomial fit and ULI for the monthly ($R^2=0.995$) and daily ($R^2=0.973$) mean values, as shown in Figure 17.

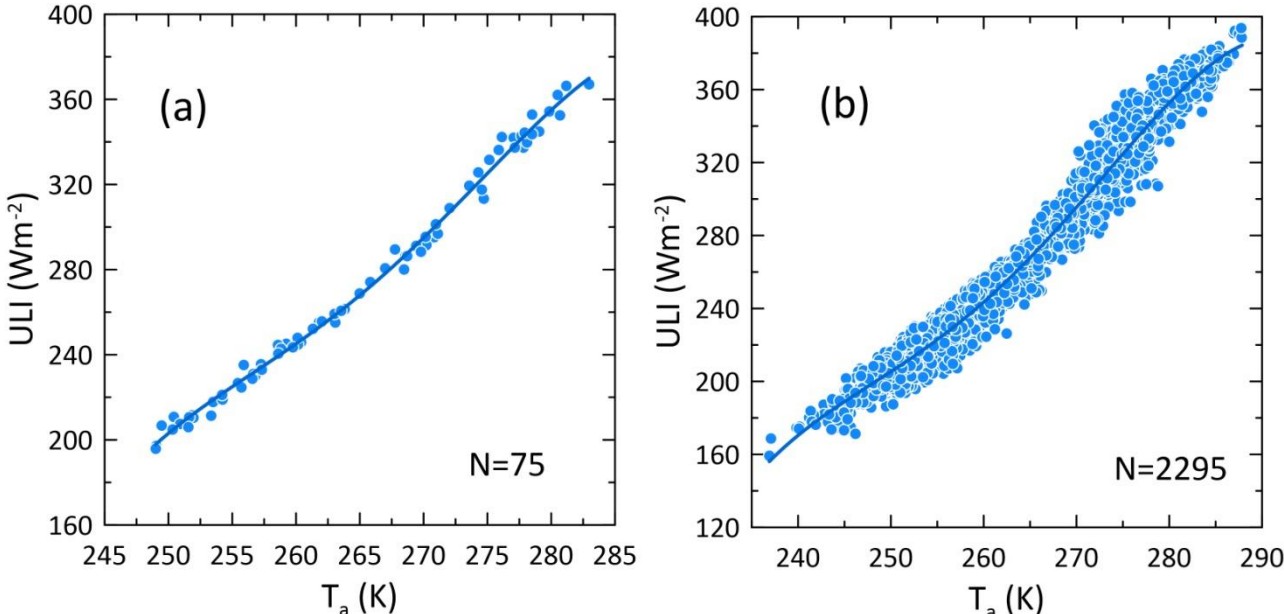

Figure 17: (a) Monthly and (b) daily correlation of $T_a$ and ULI. The curve represents the fit with a fourth-degree polynomial. The number of points used in the fit is shown in each graph.

The fourth-degree polynomials are expressed by equations (3) and (4) for the monthly and daily values, respectively:

$$ULI = -8.801 * 10^5 + 1.329 * 10^4 * T_a - 7.523 * 10 * T_a^2 + 1.891 * 10^{-1} * T_a^3 - 1.780 * 10^{-4} * T_a^4 \tag{3}$$

$$ULI = -3.624 * 10^5 + 5.594 * 10^4 * T_a - 3.233 * 10 * T_a^2 + 8.292 * 10^{-2} * T_a^3 - 7.956 * 10^{-5} * T_a^4 \tag{4}$$

### 3.3 Surface radiation budget

The Surface Radiation Budget (SRB) is calculated as the sum of the SW and LW net irradiance

$$SRB = (DLI - ULI) + (DSI - USI) = (DLI - ULI) + DSI * (1 - A) \tag{5}$$

The annual distribution of the monthly mean net irradiance measured at THAAO is shown in Figure 18, while monthly and seasonal means, standard deviations, minimum, and maximum values are presented in Table 9.

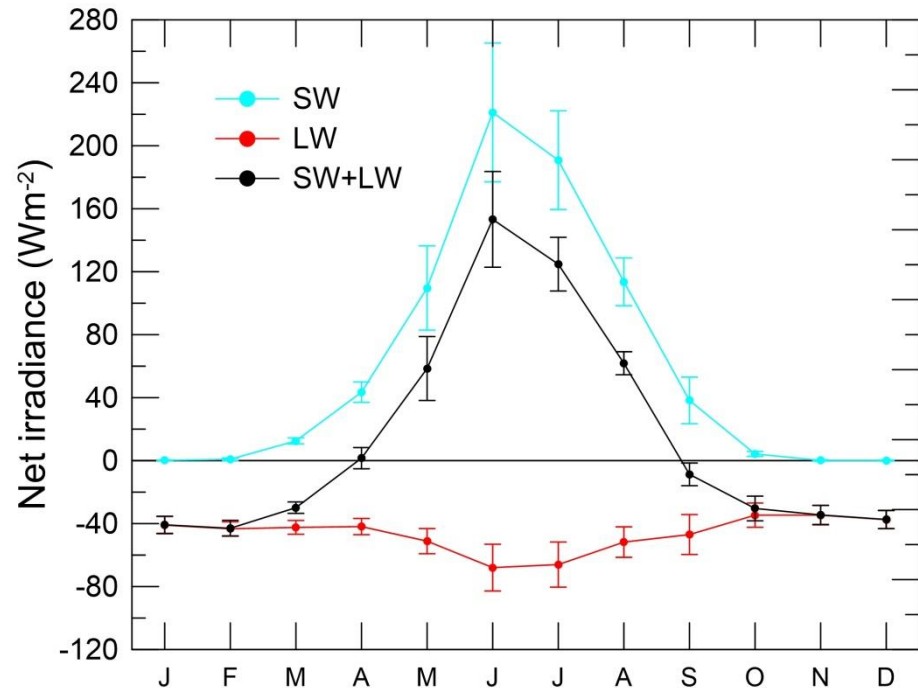


Figure 18: Monthly distribution of SW, LW, and total SRB. Vertical bars correspond to one standard deviation of the monthly averages.

Table 9. Monthly and seasonal means, standard deviation, minimum, and maximum of the net SW, LW and the SRB.
Seasonal statistics are calculated over the seasons when monthly means are available for each month.

|  | NET SW (2016-2022) | | | | NET LW (2016-2022) | | | | SRB (2016-2022) | | | |
|---|---|---|---|---|---|---|---|---|---|---|---|---|
|  | Mean | S. Dev. | Min. | Max. | Mean | S. Dev. | Min. | Max. | Mean | S. Dev. | Min. | Max. |
| JAN | 0.2 | 0.3 | -0.1 | 0.6 | -41.0 | 5.5 | -46.8 | -32.8 | -40.8 | 5.4 | -46.5 | -32.6 |
| FEB | 0.9 | 0.8 | -0.6 | 1.4 | -43.3 | 4.4 | -48.7 | -38.5 | -43.0 | 5.0 | -47.6 | -37.4 |
| MAR | 12.5 | 1.9 | 10.3 | 14.7 | -42.4 | 4.4 | -47.6 | -36.6 | -29.9 | 3.6 | -32.9 | -23.8 |
| APR | 43.5 | 6.5 | 35.8 | 51.3 | -41.9 | 5.2 | -47.9 | -33.7 | 1.6 | 6.7 | -5.2 | 10.2 |
| MAY | 109.6 | 26.8 | 77.0 | 133.0 | -51.1 | 8.0 | -60.3 | -40.9 | 58.5 | 20.3 | 36.1 | 80.6 |
| JUN | 221.2 | 44.1 | 164.2 | 275.7 | -67.9 | 14.8 | -83.1 | -45.5 | 153.2 | 30.4 | 118.7 | 196.0 |
| JUL | 190.8 | 31.3 | 158.5 | 229.3 | -66.0 | 14.3 | -83.6 | -51.2 | 124.8 | 17.1 | 107.0 | 145.7 |
| AUG | 113.6 | 15.2 | 91.2 | 134.1 | -51.7 | 9.7 | -64.1 | -40.0 | 61.9 | 7.3 | 49.8 | 70.0 |
| SEP | 38.2 | 14.8 | 20.4 | 63.9 | -47.0 | 12.7 | -65.3 | -33.0 | -8.7 | 7.2 | -16.9 | 1.5 |

| | | | | | | | | | | | | |
|---|---|---|---|---|---|---|---|---|---|---|---|---|
| **OCT** | 4.3 | 1.5 | 2.4 | 6.9 | -34.6 | 7.7 | -48.5 | -25.6 | -30.4 | 7.8 | -45.5 | -23.2 |
| **NOV** | 0.1 | 0.4 | -0.2 | 0.7 | -34.6 | 6.2 | -43.1 | -26.9 | -34.5 | 6.1 | -42.3 | -26.8 |
| **DEC** | 0.1 | 0.3 | -0.3 | 0.6 | -37.5 | 5.7 | -43.0 | -29.9 | -37.4 | 5.8 | -42.5 | -29.7 |
| | | | | | | | | | | | | |
| **MAM** | 55.8 | 11.2 | 43.6 | 66.3 | -45.9 | 2.1 | -47.9 | -43.9 | 9.9 | 9.1 | -0.2 | 18.8 |
| **JJA** | 174.1 | 23.4 | 154.0 | 204.9 | -61.1 | 9.6 | -71.8 | -52.2 | 113.0 | 14.3 | 100.2 | 133.1 |
| **SON** | 14.2 | 5.0 | 8.0 | 22.5 | -38.4 | 2.2 | -40.7 | -34.6 | -24.2 | 4.1 | -29.3 | -17.8 |
| **DJF** | 0.3 | 0.4 | -0.2 | 0.5 | -40.4 | 4.3 | -44.8 | -33.7 | -40.0 | 4.4 | -44.3 | -33.2 |

The net SW and LW peak in June (221.2 Wm$^{-2}$ and -67.9 Wm$^{-2}$, respectively), with the solar component larger than the infrared one. The shape of the SW net irradiance is not symmetric with respect to the summer solstice, with spring values lower than autumn ones: this is due to the high albedo values in spring, causing lower SW net irradiances in this season.

The total SRB is positive from April to August; during this period, the solar heating at the surface exceeds the infrared cooling. The net surface irradiance displays a maximum in June of 153.2 Wm$^{-2}$.

From October to March the SRB is negative, as the outgoing LW irradiance is larger than the incoming one, and the SW component is not present. The net LW radiation varies between -34.6 and -43.3 Wm$^{-2}$. In March and September, the absolute value of the net LW irradiance is larger than the net SW component, resulting again in a negative SRB.

Previous studies on the SRB in Greenland are scarce, and most of them address all the components of the surface energy budget. Miller et al. (2015) and Miller et al. (2017) present, respectively, nearly three years (January 2011- October 2013) and one year (July 2013 – June 2014) and of the SRB cycle at Summit Station (72° N, 38° W, 3211 m), on the GrIS. The most evident peculiarity of the SRB over the GrIS is the all-year-round high USI, due to the permanent high albedo conditions. Being at a lower latitude than THAAO, the DSI monthly means are larger, but the resulting net SW irradiance is

much lower, with a peak of 60 Wm$^{-2}$ in June. The combined effect and altitude and lower temperatures determine DLI and ULI data below those at THAAO. The net LW irradiance has winter values of -20 Wm$^{-2}$ and a maximum intensity in June of about -50 Wm$^{-2}$ (Miller et al., 2015; 2017), resulting in a negative SRB throughout the year, excluding June and July.

Observations of SRB in five coastal sites in Greenland are available in 2012-2015 (Lund et al., 2017). Three sites are located in North-East Greenland near the Zackenberg Research Station (latitudes 74.5-74.6° N, longitudes 20.6-21.4° W) and are

representative of wet and dry tundra and the glacier ablation area, while two are South-West Greenland in the Nuuk area (latitudes 64.1-64.5° N, longitudes 49.5-51.4° W), characterized by wet tundra and the ice sheet ablation area. The tundra sites close to the Zackenberg Research Station, for which measurements in 2012 and 2013 are presented by Lund et al. (2017), have a similar timing of snow melt to THAAO, causing the decrease of surface albedo in June to values generally between 0.1 and 0.2 and the differences in the net fluxes before and after snow melt. Contrarily to the rocky surface at

THAAO, the tundra may heat up to temperatures around 20-30 °C, much larger than those reached at THAAO, and the net

LW irradiance may differ. Lund et al. (2017) do not provide monthly means to be compared with the numbers of the present study.

Measurements of SRB in the Arctic for extended periods have been carried out in Alaska, at the NOAA Barrow Observatory in Utqiaġvik and at the close ARM North Slope of Alaska site (Dong et al., 2010), at Ny-Ålesund, Svalbard (Maturilli et al.,
2015), at Eureka, Canadian Arctic Archipelago in Nunavut, and at Tiksi, East Siberia (Grachev et al., 2018), as well as in the Beaufort and Chukchi Seas North of Alaska during the SHEBA field experiment (Persson et al., 2002; Intieri et al., 2002; Shupe and Intieri, 2004).

Measurements during the 1997-1998 SHEBA experiment show a positive net irradiance from May to August, with maxima of 80 Wm$^{-2}$ in July (Persson et al., 2002): however, during the one year-long campaign, the latitude of the ice floe shifted
from 74° to 81° N, and this may have influenced mainly the measured DSI.

Dong et al. (2010) present about ten years (from 1998 to 2008) of radiation, cloud, and surface parameters measurements at Barrow Observatory (71.3° N, 156.6° W). They present monthly means of DSI, USI, DLI, and ULI for all sky conditions. From their Table 1 the seasonal means of the SRB can be calculated: -28.6 Wm$^{-2}$ in winter, -0.1 Wm$^{-2}$ in spring, 103.2 Wm$^{-2}$ in summer, and -6.8 Wm$^{-2}$ in autumn.

The 21 year-long record (1992-2013) of SRB at Ny-Ålesund (78.9° N, 11.9° E), Svalbard, is presented by Maturilli et al. (2015). The annual cycle shows positive SRB from May to August, with a peak in July around 100 Wm$^{-2}$, lower than the values obtained at THAAO. The different latitude at the two sites determines lower SW maxima at Ny-Ålesund. In addition, the net LW minimum at Svalbard, around -50 Wm$^{-2}$, is less intense than at THAAO.

Eureka is the site closest to THAAO in terms of distance. Grachev et al. (2018) present annual cycles of the surface fluxes
and other ancillary data using hourly data. The SRB has similar timing compared with THAAO, although the LW component has lower intensity (as absolute values). Similarities in the changes in surface albedo, occurring in May and September, are also found; however, summer albedo values are larger at Eureka, especially in July.

Differences in radiation regimes among sites can be ascribed to many factors, such as meteorological conditions, latitude, cloud properties, and surface type.


## 4 Data availability

The time series of $T_a$ (Muscari et al., 2018; https://doi.org/10.13127/thaao/met), DSI (Meloni et al., 2022a; https://doi.org/10.13127/thaao/dsi), USI (Meloni et al., 2022b; https://doi.org/10.13127/thaao/usi), DLI (Meloni et al., 2022c; https://doi.org/10.13127/thaao/dli), and ULI (Meloni et al., 2022d; https://doi.org/10.13127/thaao/uli) can be visualized and
downloaded through the THAAO web site (https://www.thuleatmos-it.it/data).

# 5 Conclusions

The Thule High Arctic Atmospheric Observatory (THAAO, 76.5° N, 68.8° W), in North-Western Greenland, is a long-term measurement site of atmospheric composition and upper air vertical profiles, where the downward components of the shortwave and longwave irradiances (DSI and DLI) are measured since 2009, while the upward components (USI and ULI) since 2016.

The four components are measured with regularly calibrated instruments to monitor the evolution of their sensitivity with time.

The pyranometers are characterized in terms of thermal offset and the Eppley model PSP, measuring the DSI also for the cosine response. For the Kipp&Zonen radiometers, whose temperature dependence of the sensitivity constant is determined, correction to the data is applied accordingly. The assumed expanded uncertainty resulting from the calibration of the pyranometers is ±1.41% for DSI from the CMP21 and ±2% for DSI and USI from the PSP, while is ±5 Wm$^{-2}$ for DLI and ULI.

The BSRN recommended quality checks are applied and the data outside the test boundaries for the "extremely rare limits" are rejected in this analysis.

The original datasets can be visualized and accessed as hourly means at the THAAO web site (see https://www.thuleatmos-it.it/data/index.php), while in this study daily and monthly means are presented and discussed. The evolution of daily and monthly air temperature data since July 2016 is also presented, as this parameter is strongly linked to DLI, ULI, and surface albedo, as discussed in the following.

The air temperature record at THAAO captured some interesting synoptic events, such as the anomalous cold period of January-March 2020, consequent to the exceptionally strong and cold stratospheric polar vortex in the Northern Hemisphere in winter 2019-2020, and the warmest summer 2019, that caused a significant Greenland Ice Sheet (GrIS) mass loss.

DSI peaks in June (mean 277.0 Wm$^{-2}$), when the SZA reaches its minimum value at 53°, while is absent from 29 October to 13 February (SZA≥90°). USI is not present as well in the same period, but its peak is anticipated to May (132.4 Wm$^{-2}$) due to the persistence of snow/ice on the ground during this month, when DSI has already reached high values.

The USI/DSI ratio provides the shortwave surface albedo (A). During summer (from mid-June to the end of August) the surface is free from snow and the surface albedo varies between 0.13 and 0.18, with an average of 0.16 and little interannual variability. During months of snow-covered surface, when solar radiation allows to estimate A, its values are >0.6. The end of May and September are generally characterized by rapidly changing surface conditions, although with marked interannual variability, due to differences in the onset/conclusion of the snowfall season or transport/removal from wind or air temperatures. For example, the period from mid-April to May 2021 was characterized by A values lower than average, that was triggered by an event of steep increase in air temperature, characterized by values above the melting point and a large positive anomaly (calculated with respect to the 2016-2022 average) > 11 K that possibly caused an anticipated snow melt. This hypothesis is in line with the longwave fluxes of May 2021, respectively the largest of the DLI record (from 2016 to

2022) and the second largest of the ULI record. Similarly, a period of large $T_a$ and positive $T_a$ anomalies with maxima of 7-8 degrees from the end of August to the end of September 2019, may be responsible for persistent snow-free conditions in this month: in particular $T_a$ values remain above the melting point for six days, with a peak of 280 K on 28 September corresponding to about 9 K of $T_a$ anomaly.

DLI has maxima values in July and August (mean values of 291.2 $Wm^{-2}$ and 288.7 $Wm^{-2}$, respectively), and minima in February and March (mean values of 172.5 $Wm^{-2}$ and 174.4 $Wm^{-2}$, respectively), while ULI is always larger than DLI, and with a wider annual cycle peaking with a maximum in July (356.1 $Wm^{-2}$) and a minimum in February and March (210.3 $Wm^{-2}$ and 211.6 $Wm^{-2}$). During the polar night, DLI and ULI are the only flux components, with averages over the period November-January of 192.6 $Wm^{-2}$ and 231.8 $Wm^{-2}$, respectively. Generally, May and June are the months when DSI reaches at least 50% of the total irradiance.

ULI and $T_a$ show a similar time evolution. ULI appears to be well described by a four-order polynomial curve in $T_a$.

The surface radiation budget has been calculated from the THAAO monthly averaged flux components. The SRB is positive from April to August, when solar heating exceeds infrared cooling, with a maximum of 153.2 $Wm^{-2}$ in June. From November to February, during the polar night, the SRB is mostly negative, due to the larger outgoing LW irradiance compared to the incoming one, and the net LW irradiance varies between -34.5 and -43.0 $Wm^{-2}$. In March and September, the LW overcomes the SW component resulting in negative SRB. The largest infrared cooling at the surface is measured in June (-67.9 $Wm^{-2}$) and July (-66.0 $Wm^{-2}$).

Direct observations of the surface radiative fluxes and of the radiation budget at THAAO represent a valuable contribution to studying the Arctic climate. Indeed, THAAO is a coastal site facing the sea and the terminus of ice streams of the Wolstenholme Fjord, both undergoing changes related to rapid temperature increase. THAAO measurements add to those of very few Arctic sites providing long-term records of the surface radiation budget, required to validate satellite products, reanalyses, and regional and global climate models. In addition, such measurements allow the evaluation of the role of aerosol and clouds and their radiative effects in the evolution of the Arctic climate.

**Author contributions**

DM analysed the data, prepared the datasets, and wrote the paper. FCQ, VC, ADB, TDI, GP, CS maintained the instruments at THAAO, in particular the radiometers. GM analysed the data, managed the instruments, in particular the meteorological station. AI produced and managed the THAAO web site and the data archiving and sharing service. AdS installed the radiometers and made intercalibrations, supervised the analysis and the paper design. All the authors were involved in the review and editing process.

**Competing interests**

The authors declare that they have no conflict of interest.

**Acknowledgements**

Authors acknowledge the United State Space Force and Pituffik Space Base for hosting THAAO. In particular, the collaboration of Charles Klinger, Jeffrey Villa, and Otto Lee in the management of the instruments at THAAO is greatly appreciated. We thank the National Science Foundation for providing services at THAAO and the Polar Field Services for the logistic support.

The contribution of dr Claudia Di Biagio to the initial radiation measurements at THAAO is gratefully acknowledged.

We acknowledge the use of imagery from the NASA Worldview application (https://worldview.earthdata.nasa.gov/), part of the NASA Earth Observing System Data and Information System (EOSDIS).

The authors wish to acknowledge the two anonymous reviewers for their comments that helped to improve the manuscript.

**Financial support**

This research has been supported by the Italian Antarctic research program (PNRA) through the Clouds And Radiation in the Arctic and Antarctica (CLARA2) project, and by the Italian Arctic Research Programme (PRA) through the Effects of Changing Albedo and Precipitation on the Arctic Climate (ECAPAC) project. Both PNRA and PRA are directed by the Ministry of University and Research. The Italian activities at THAAO have been supported by the Istituto Nazionale di Geofisica e Vulcanologia (Environment Department) in the framework of the Multidisciplinary Analysis of Climate change indicators in the Mediterranean And Polar regions (MACMAP) project.

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
