# Peer review of "Shortwave and longwave components of the surface radiation budget measured at the Thule High Arctic Atmospheric Observatory, Northern Greenland"

_Earth System Science Data, 2023_

## Author Comment (AC1)

Referee #1

The validation and assessment of surface radiation flux products derived from satellite observations is difficult, especially in the Arctic region, where there are fewer observing stations. This study presents daily and monthly surface radiative fluxes (SRF) data collected at the Thule High Arctic Atmospheric Observatory (THAAO, 76.5° N, 68.8° W), in North-Western Greenland. These data are essential for the validation of satellite data. In the current state of the manuscript, some doubts need to be addressed.

1.  I think it will be of interest to some readers to know why the THAAO was established at 76.5°N, 68.8°W. The addition of a diagram of the geographic location of the THAAO would also be helpful for reading.

    A detailed description of the history and scientific purposes of the THAAO is provided in Muscari et al. (2014), whose citation has been added to the manuscript. A summary of the scientific motivation around THAAO has been added in Section 2.1: "The collaboration between DMI, at that time already involved with measures dedicated to the study of the Arctic climate at the Thule Air Base, and the Italian institutions was born with the installation of a Rayleigh lidar by the University of Rome in addition to the DMI instruments (ozonesondes and UV/visible spectrometer), to improve knowledge of the stratospheric ozone depletion phenomenon that was observed to be very intense over Antarctica and anticipated to become important also over the Arctic. (Muscari et al., 2014)".

    A combination of two Google Earth images has been used to create the new Figure 1 to highlight the position of the THAAO.

2.  Figure 3 shows the variation of the ratio of DSI with solar zenith angle for the spring and summer of 2012 for both instruments. What is the reason for selecting the data for 2012, and what is the variation of the ratio of DLI, USI, and ULI with solar zenith angle? What are the reasons for the large fluctuations in the ratio of the two observations at solar azimuths of 70 to 80?

    Figure 3 (Figure 4 of the revised paper) shows how to derive the cosine response of the Eppley PSP pyranometer, in particular of the instrument with sn 34891F3, which is not determined by the manufacturer. In this case the PSP was deployed for some days close to CMP21 sn 090206, with the double aim of checking the PSP sensitivity and deriving its cosine response. The Kipp&Zonen CMP21 has a good cosine response, which is measured at the factory and is below 0.7% up to 60° zenith angle and below 1.2% up to 80°, so it is taken as reference. The intercomparison was carried out at the Climate Observatory in Lampedusa Island, in the Mediterranean South of Sicily, where frequent cloud-free conditions occur, in 2012, after the PSP34891F3 was installed at THAAO. Once determined, the DSI measured by the PSP sn 34891F3 before and after 2012 was corrected for the derived cosine response as a function of the solar zenith angle.

    The reason for the large fluctuations at SZA of 70-80° is the low signal produced when the sun is close to the horizon: in this case small differences in the measured signal lead to large ratios. In addition, the presence of aerosols or thin clouds near the horizon may determine inhomogeneities in the signals.

    USI depends on SZA, but since it is a measure of the radiation that is reflected by the surface it is only diffuse, which is considered to be nearly isotropic and not directional, as the direct component is. The variation of the ratio of USI with SZA has not been determined, and appears to be negligible because the upward component of the solar irradiance is affected in a complex way which depends on the surface properties. For instance, the dependence on SZA is zero for a lambertian surface.

DLI and ULI do not directly depend on SZA, but manly on temperature and emissivity of the sky and the terrain, respectively.

3.  What is the typical polar environment referred to in line 334? Is the environment in the area where the observatory is located different from other regions of the Arctic? What are the reasons for the largest differences in DLI in this environment? Also, the difference in Figure 4(b), where day of year is 78, appears to be uncorrected.

    The sentence refers to the polar regions in general, where DLI may reach values as low as 120-130 Wm$^{-2}$. These values are typical of an environment characterized by very low temperatures and column water vapour, i.e., in polar regions or at very high altitude sites, like Himalaya. DLI values below 120 Wm$^{-2}$ can be measured on the Greenland Ice Sheet, due a combination of altitude and geophysical parameters.

    The differences shown in Figure 4 (Figure 5 of the revised paper) for low DLI values are attributed to the fact that both PIR and CGR4 pyrgeometers calibration is performed for DLI values typical of mid latitudes. Moreover, the CGR4 model has the temperature dependence of the sensitivity determined down to -40 °C, while the PIR model sensitivity is independent of the instrument temperature. The combination of these factors is responsible for the differences between the measurements of the two instruments at low temperatures (i.e., during Arctic winter). A sentence is added in the text to highlight this aspect.

    Some data in Figure 4b reasonably appear as uncorrected: they correspond to measured DLI values above 270-280 Wm$^{-2}$, for which the application of the original PIR coefficients and the new ones gives very similar values (see figure 4a). This is the case for measurements on days 74, 77, and 80.

4.  Does the phrase "extremely rare limits" in line 358 include limits due to meteorological conditions?

    No, it does not include meteorological conditions but it refers to irradiance values only. According to BSRN definitions, the "physically possible limits" have been fixed considering the SW and LW irradiance values given the wide range of meteorological conditions found all over the entire globe. The "extremely rare limits" are limits for minimum acceptable values for SW irradiance measurements based on the fact that values below -4 Wm$^{-2}$ may reflect a not proper correction of the thermal offset, thus those data quality may not be acceptable (Long and Shi, 2008).

5.  What is the Pearson correlation coefficient between temperature and radiative flux in figure 5?

    The Pearson correlation coefficient is 0.82 for DLI and 0.97 for ULI.

6.  In line 388, what is the sample size below which the sample size is too small? Could the authors please provide information on the sample size of valid observations during the observation period?

    The minimum sample sizes required to calculate means are 45 minutes, 18 hours, and 22 days, respectively, for hourly, daily, and monthly means.

    Following the reviewer suggestion, we calculated the percent of valid observations per year for the dataset available for download, i.e hourly means. The results are presented in Table 6 of the revised manuscript.

7.  Lines 405-420 refer to temperature anomalies, and it is suggested that the time-series variation of temperature anomalies be presented in a graph. Have the temperature observations been validated against NCEP reanalysis information?

We agree with the reviewer to add the graph showing the time series of daily and monthly temperature anomalies. The graph is added as Figure 7b and some comments to the figure are added in the text.

Similarly, we did the same calculating the DSI, USI, DLI, and ULI anomalies, adding and commenting the corresponding figures in Sections 3.2.1 and 3.2.3. In particular, USI anomalies are related to surface melting, while DLI and ULI anomalies to air temperature anomalies.

The monthly mean THAAO air temperature has been compared to NCEP reanalysis to answer to the reviewer question. Data from NCEP-DOE reanalysis 2 dataset have been used (Kanamitsu et al., 2002), in particular the monthly means of the air temperature at 2 m, which is directly comparable to our dataset. The resolution of the dataset is 2.5°x2.5° and we did expect a poor correlation between THAAO measurements and model data, due to position of the observatory, being along the coast, close either to the sea and to the ice sheet, thus representative of local conditions. The results of the comparison confirm our premises. The four grid point closest to THAAO are shown in the figure: NCEP_1 (75.235° N,67.5° W), NCEP_2 (77.139° N,67.5° W), NCEP_3 (75.235° N,69.375° W), NCEP_4 (77.139° N,69.375° W).

[Figure]

The time series of $T_{air}$ for THAAO and for the four NCEP grid points is plotted in the following figure.

[Figure]

The curves relative to Tair for points NCEP_1 and NCEP_3, which are indeed relatively close one to the other, are nearly superimposed, and the same is for points NCEP_2 and NCEP_4. NCEP_1 and NCEP_3 are on the sea, so we concentrated on data from NCEP_4, which is the closest to THAAO and also closest to the coast, similarly to THAAO. THAAO measurements are always above the reanalysis, as also shown by the scatterplot with the linear regression. The average difference is (-7.1±1.9) K.

[Figure]

The most plausible explanation for these results is that the resolution of the NCEP reanalysis is not adequate for comparison with the THAAO data due to the location of the measurement site close to the coast, and therefore with surface and atmospheric characteristics (sea at about 2 km and glacier about 15km) very different within a short distance.

Given the obtained results, we compared observations with ERA5 reanalysis (Hersbach et al., 2023), which has a much finer spatial resolution (31 km) than NCEP and also provides monthly mean temperature data at pressure levels. Since the average pressure value in 2016-2022 at THAAO is 982 mbar, we present the results for 1000 mbar and 975 mbar.

[Figure]

The temporal evolution of THAAO temperature is very well reproduced by the ERA5 data, although $T_a$ at 1000 mbar better reproduce the larger values, those occurring from April to September, within $\pm 1$ K, while overestimating winter temperatures, up to 3.8 K in February.

[Figure]

$T_a$ at 975 mbar are lower than those at 1000 mbar, determining a better agreement in winter, with differences decreasing to 2 K in February.

Generally speaking, reanalysis data over the Arctic are unlikely to be well constrained by the poor and discontinuous existing observations (e.g. Orsi et al., 2017).

A warm bias in ERA5 reanalysis relative to buoy observations over the Arctic sea ice has been documented, with smaller bias in summer and larger in winter (e.g. Wang et al., 2019). The warm

bias is attributed to the poor ability of the reanalysis to simulate the atmosphere/sea ice interaction (e.g. Batrak and Müller, 2019), the shallow Arctic boundary layer and the presence of intense temperature inversions (e.g. Tjernström and Graversen, 2009).

Since the comparison of THAAO air temperature and reanalysis products is beyond the scope of the paper, it has not been included in the revised manuscript.

References

Batrak, Y., Müller, M. On the warm bias in atmospheric reanalyses induced by the missing snow over Arctic sea-ice. Nat Commun 10, 4170 (2019). https://doi.org/10.1038/s41467-019-11975-3.

Hersbach, H., Bell, B., Berrisford, P., Biavati, G., Horányi, A., Muñoz Sabater, J., Nicolas, J., Peubey, C., Radu, R., Rozum, I., Schepers, D., Simmons, A., Soci, C., Dee, D., Thépaut, J-N. (2023): ERA5 monthly averaged data on pressure levels from 1940 to present. Copernicus Climate Change Service (C3S) Climate Data Store (CDS), DOI: 10.24381/cds.6860a573 (Accessed on 21 September 2023).

Kanamitsu, M., Ebisuzaki, W., Woollen, J.S., Yang, S., Hnilo, J.J., Fiorino, M., & Potter, G. (2002). NCEP–DOE AMIP-II Reanalysis (R-2). Bulletin of the American Meteorological Society, 83, 1631-1643. (Accessed on 19 September 2023).

Orsi, A. J., Kawamura, K., Masson-Delmotte, V., Fettweis, X., Box, J. E., Dahl-Jensen, D., Clow, G. D., Landais, A., and Severinghaus, J. P.: The recent warming trend in North Greenland, Geophys. Res. Lett., 44, 6235–6243, doi:10.1002/2016GL072212, 2017.

Tjernström, M., and R. G. Graversen, 2009: The vertical structure of the lower Arctic troposphere analysed from observations and the ERA-40 reanalysis. Quart. J. Roy. Meteor. Soc., 135, 431–443, https://doi.org/10.1002/qj.380.

Wang, C., Graham, R. M., Wang, K., Gerland, S., and Granskog, M. A.: Comparison of ERA5 and ERA-Interim near-surface air temperature, snowfall and precipitation over Arctic sea ice: effects on sea ice thermodynamics and evolution, The Cryosphere, 13, 1661–1679, https://doi.org/10.5194/tc-13-1661-2019, 2019.

Acknowledgements

NCEP/DOE Reanalysis II data provided by the NOAA PSL, Boulder, Colorado, USA, from their website at https://psl.noaa.gov.

Hersbach, H. et al., (2023) was downloaded from the Copernicus Climate Change Service (C3S) (2023).

The results contain modified Copernicus Climate Change Service information 2020. Neither the European Commission nor ECMWF is responsible for any use that may be made of the Copernicus information or data it contains.

8. Does the maximum and minimum of the observed data refer to the average result of the observed maximum and minimum?

No, we did not analyze the observed minimum and maximum temperature. $T_a$ min and $T_a$ max in Table 6 represent the smallest and largest monthly and seasonal means for $T_a$. This has been explained in the text.

9. As global warming occurs, will the area and duration of snow cover in the Arctic decrease? Does this have an effect on the observation of shortwave radiative fluxes?

   The progressive decrease of sea ice and snow cover is well documented (e.g., IPCC, 2019). This is expected to largely impact shortwave radiation, and in particular USI (but also ULI). Due to the large interannual variability of these parameters it is not possible to derive a trend from the THAAO dataset.

10. Is there a relationship between the monthly average expression of long-wave radiative fluxes and cloudiness? If so, what is the correlation between them?

    DLI is strongly affected by the presence of clouds, which produce an enhancement of DLI compared to cloud-free conditions. The amount of this increase is linked to atmospheric (air temperature, integrated water vapour) and cloud (cloud cover fraction, cloud base altitude, liquid water or ice path, cloud optical thickness, cloud droplet size). We are preparing a paper describing the cloud radiative effects on both the shortwave and longwave at THAAO since 2016, and its dependence on atmospheric thermodynamics, cloud base altitude (i.e., temperature), liquid water path, cloud optical thickness. Our preliminary results show an increase of the longwave cloud radiative effect as a function of air temperature, liquid water path, cloud base temperature.

11. What is the trend of the longwave and shortwave radiative fluxes over the observed time period? Are the trends due to meteorological conditions or changes in the surface type?

    The calculation of the trend of the radiative flux components is not part of this work as the length of the time series would not allow statistically robust results, in particular due to the large interannual variability of the involved processes. Furthermore, the available time series meteorological parameters (since 2016) would not allow us to investigate the causes of any detected trends.

12. The paper has relatively little comparison and discussion with other similar studies known in the field. Can you provide more details about how your measurements compare to existing similar studies, highlighting where this dataset fills a gap?

    There is unanimous consensus among the scientific community that the climate change taking place globally is particularly intense and rapid in the Arctic. Many complex and interconnected phenomena operate in this region and need to be understood in order to be able to predict the evolution of the climate with a good level of accuracy. To do this, it is necessary to start from the observation of phenomena on the longest time scales. Due to the difficult climatic conditions, it is complex to maintain observation sites over long periods in the Arctic. This is why having data from an observatory that monitors changes in many parameters of climatic interest (not only meteorological parameters, but also radiative budget, aerosols, clouds, trace gases) is of paramount relevance.

    The paper by Uttal et al. (2016) highlights the importance of combining data from different observatories in the Arctic, grouped under the International Arctic Systems for Observing the Atmosphere (IASOA) umbrella, to also cope with the diversity of geographical and climatic conditions. Regarding radiation measurements, specifically, figure 5 of the article shows the differences in the annual evolution of the surface albedo for 4 sites of the BSRN network belonging

to IASOA. The curves reflect a different snowfall timing of the transition from snow-covered to snow-free conditions for each site, as it is linked to various local factors, such as snow accumulation, temperature, and cloudiness.

The THAAO observations add to the very few available in the Arctic, useful for investigating climate variability on a smaller scale than that which includes the entire Arctic. Established Arctic observatories close to the coast with long term radiation fluxes measurements are Ny-Ålesund, Barrow, Eureka, Tiksi. THAAO has the peculiarity of being close to both the sea and the Greenland Ice Sheet, thus being influenced by different surface and atmospheric conditions.

The paper by Uttal et al. (2016) has been cited in the introduction and in Section 3.2.2.

Could you provide a more detailed description of the procedures associated with the processing of the original data, e.g., procedures for data quality control, uncertainty estimation, etc., to improve the accuracy and reliability of the study.

We provide a detailed description of the SW irradiance data processing in Section 2.1.2, and the procedures to reduce measurement uncertainties (thermal offset correction, calibration checks and cosine response correction) in Sections 2.1.3 and 2.1.4, while Section 2.1.5 describes the pyrgeometers calibration. The quality control process is described in Section 2.1.6.

A sentence was added in lines 245-247 to describe the formulas used to calculate irradiance from the raw datalogger signals.

13. In the introduction, the authors mention that ground-based measurements can improve the discrepancy of climate models in modeling Arctic warming. Have you considered comparing measurements with simulations from existing models? Provide an in-depth analysis of the importance of existing measurements for model simulation assessment.

Yes, we are currently coordinating a national project, "Effects of changing albedo and precipitation on the Arctic climate" (ECAPAC), focused on quantifying the impact of precipitation on the surface distribution of ice and snow and on the surface radiation budget. One of the activities within the project is to assess the performance of a regional model specifically developed for polar regions, the Modèle Atmosphérique Régionale (MAR) (Fettweis et al., 2017; Tedesco et al., 2016), over North-Western Greenland, a region where it has not been tested yet. The assessment is mainly aimed at quantifying the contribution from atmospheric properties and land-ice components to the model uncertainties and to investigate the effect of assimilating into the model measured parameters such as precipitation (amounts and phase), water vapor concentration profiles and surface radiation fluxes. In addition, it would be possible to directly compare the measured radiative fluxes with the MAR output.

As also shown by the comparison of THAAO measurements with meteorological reanalysis, global climate models or reanalysis are optimized to provide a good representation of the global mean climate and their spatial grid, including the orography description, is necessarily large. On the contrary, regional climate models run over a limited area, with finer spatial resolution and orography, and can be calibrated to represent the physical processes occurring in a specific region. For these reasons, regional models can reduce the biases found in global models or reanalysis.

The importance of measurements in model assessment is twofold. On one side, observations are used during model development, calibration and inizialization, but also to tune the model in order to match observations. Thus, the quality of the model products depends on the quality of the underlying observations. Similarly, the performance of a model is assessed by comparison against

measured data records. Consequently, the uncertainty on the model product is influenced by the uncertainty on the observational reference (e. g. Kotlarski et al., 2017).

Kay et al. (2016) emphasize how the advances in Arctic cloud and climate research in the past benefited from the combination of modeling and observations, including those from satellites with active remote sensing instruments (CloudSat and CALIPSO), and that the best way forward is to exploit recent satellite missions, but also in situ observations over long periods, and intensive measurement campaigns, conducted with instruments on various platforms.

Past Arctic field campaigns, like the Surface Heat Budget of the Arctic campaign, SHEBA (Uttal et al., 2002), a year-long drift experiment that took place in the Beaufort and Chukchi Seas from October 1997 to October 1998, allowed to identify wintertime biases in regional climate models (e.g. Wyser et al., 2008) and in global forecast systems (e.g. Simjanovski et al., 2011). In both cases surface albedo and cloud representations are the main reasons for the estimated biases.

The Norwegian young sea Ice cruise, N-ICE2015, was conducted in the Arctic Ocean north of Svalbard from January to June 2015 (Walden et al., 2017). Di Biagio et al. (2021) used meteorological and radiation measurements, as well as cloud lidar observations to derive biases in ERA5 and ERA-Interim reanalysis and in CERES surface radiation fluxes.

Recently, the Multidisciplinary drifting Observatory for the Study of Arctic Climate (MOSAiC) expedition from October 2019 to September 2020 offered the opportunity to characterize all aspects of the Arctic atmospheric system in unprecedented detail, using a variety of approaches, and across multiple scales (Shupe et al., 2022). Simulations of wintertime surface energy budget with short-term forecasts from 7 state-of-the-art operational and experimental forecast systems reveal that generally model struggle to represent thin liquid clouds in the Arctic which impact on longwave fluxes (Solomon et al., 2023).

An example of validation of satellite and model SRF using ground-based measurements in the Arctic is from Wang et al. (2021). They used the measured components of the SRB from four independent ground observation networks (BSRN, CEOP, TPDC, and NMC) and compared them with daily and monthly data from CERES and from various reanalysis, among which MERRA-2, NCEP-CFSR, ERA5.

We believe that an in-depth analysis of the importance of existing measurements for model simulation assessment is beyond the scope of the paper. We reported a synthesis of the cited studies in the Introduction.

References

Di Biagio, C., Pelon, J., Blanchard, Y., Loyer, L., Hudson, S. R., Walden, V. P., Raut, J. –C., Kato, S., Mariage, V., and Granskog, M. A.: Toward a better surface radiation budget analysis over sea ice in the high Arctic Ocean: a comparative study between satellite, reanalysis, and local-scale observations. J. Geophys. Res. Atmos., 126, e2020JD032555, doi:10.1029/2020JD032555, 2021.

Fettweis, X., Box, J., Agosta, C., Amory, C., Kittel, C., Lang, C., van As, D., Machguth, H. and Gallée, H., 2017, Reconstructions of the 1900--2015 Greenland ice sheet surface mass balance using the regional climate MAR model, The Cryosphere, 11, 1015–1033, https://doi.org/10.5194/tc-11-1015-2017.

Kotlarski S, Szabó P, Herrera S, et al. Observational uncertainty and regional climate model evaluation: A pan-European perspective, Int J Climatol., 2019, 39:3730–3749, https://doi.org/10.1002/joc.5249.

Shupe, MD, et al. 2022. Overview of the MOSAiC expedition: Atmosphere. Elem Sci Anth, 10: 1. DOI: https://doi.org/10.1525/elementa.2021.00060.

Simjanovski, D, Girard, E, Du, P. 2011. An evaluation of Arctic cloud and radiation processes simulated by the limited-area version of the global multiscale environmental model (GEM-LAM). Atmosphere-Ocean 49(3): 219–234. DOI: http://dx.doi.org/10.1080/07055900.2011.604266.

Solomon, A, Shupe, MD, Svensson, G, Barton, NP, Batrak, Y, Bazile, E, Day, JJ, Doyle, JD, Frank, HP, Keeley, S, Remes,T, Tolstykh, M. 2023.The winter central Arctic surface energy budget: A model evaluation using observations from the MOSAiC campaign. Elem Sci Anth, 11(1). DOI: https://doi.org/10.1525/elementa.2022.00104.

Tedesco, M., Doherty, S., Fettweis, X., Alexander, P., Jeyaratnam, J., and Stroeve, J., 2016, The darkening of the Greenland ice sheet: trends, drivers, and projections (1981–2100), The Cryosphere, 10, 477–496, https://doi.org/10.5194/tc-10-477-2016.

Wang, G., T. Wang, H. Xue, Validation and comparison of surface shortwave and longwave radiation products over the three poles, Int. J. Appl. Earth Obs., 104 (2021), 10.1016/j.jag.2021.102538

Wyser, K, Jones, CG, Du, P, Girard, E, Willén, U, Cassano, J, Christensen, JH, Curry, JA, Dethloff, K, Haugen, J-E, Jacob, D, Køltzow, M, Laprise, R, Lynch, A, Pfeifer, S, Rinke, A, Serreze, M, Shaw, MJ, Tjernström, M, Zagar, M. 2008. An evaluation of Arctic cloud and radiation processes during the SHEBA year: Simulation results from eight Arctic regional climate models. Climate Dynamics 30: 203–223. DOI: http://dx.doi.org/10.1007/s00382-007-0286-1

14. The introduction section needs to be strengthened.

This sentence is not very specific. Anyway, some considerations about the cloud-radiation feedback, the importance of integrated measurements in the Arctic, and the need for in situ observations to constrain model reanalysis and to assess model forecasts have been added in the introduction.

---

## Author Comment (AC2)

Referee #2

This manuscript describes short- and long-wave surface radiation data obtained at the Thule High Arctic Observatory in Northern Greenland over several years. The data are a useful contribution to surface energy budget studies in the Arctic, where according observations are scarce. The data will gain importance with the progression of the observation period.

Regarding the measurements, the authors apply very thorough corrections with regard to different sources of offsets (e.g. thermal offset due to the lack of shielding; cosine correction). The applied procedures are described in detail and easy to understand. Also, the comparison of the applied pyranometers to instruments that are calibrated at PMOD is explained, pointing to the relation of the instruments to corresponding reference devices. The manuscript is an important basis for understanding the data quality of the data set and moreover for the listing of metadata like e.g. instrument model and serial numbers. As the metadata are not included in the downloadable data set, I recommend to include the link to the manuscript on the data download page, so data users are aware of it.

I thank the reviewer for this very useful suggestion. A link to the publication has been added to the THAAO web pages for data access. See, for example, https://www.thuleatmos-it.it/dataaccess/DLI/

In the manuscript, time series of daily and monthly irradiances are shown, and single case studies presented in relation to meteorological conditions.

Overall, the manuscript is comprehensive and well written, and I recommend publication in ESSD upon addressing the comments listed below.

Comments:

L26: DSI is absent (solar zenith angle≥90°) from 29 October to 13 February. Yet, the data files partly contain data before 13 Feb, showing negative radiation values. I suggest to flag these or discard them.

Although negative DSI values are not physically possible, negative DSI values can be observed during nighttime (either among two diurnal periods and during the long polar night), due to the infrared loss of the pyranometers. Most of this effect is reduced via ventilation systems or correcting nighttime measurements with co-located pyrgeometers (as explained in Section 2.1.3 of the manuscript); nonetheless, some small negative values may still occur. The recommendation from BSRN is to keep the nighttime negatives in the archived files, "so that each individual BSRN customer can then decide how to treat the data depending on the scope and aim of their work" (Driemel et al., ESSDD, https://doi.org/10.5194/essd-2018-8). Similarly, we prefer to keep the data all year long and to advise the reader/user that the negative DSI values are artifacts due to the pyranometer's characteristics.

L118: in the 1990s

Done.

L 493: Could the the lower A values in 2021 be related to liquid precipitation with consequent solid ice formation on the ground, being darker than snow?

Liquid precipitation is one of the possible causes for the decrease in surface albedo in 2021. Other possible causes are snow melting due to rising temperature, or snow removal by wind. At that time, we did not have measurements of precipitation, but we have the "rain flag" data on the HATPRO microwave radiometer, indicating that some kind of precipitation (solid or liquid) may hit the rain sensor. The rain flag value is 0 without precipitation and becomes 1 in case of precipitation. Data are collected at about 1 minute time step. The figure shows the time series of the 5-minute averaged rain flag and A values from 15 April (day of the year 106) and 15 May (DOY 135), and the air temperature.

[Figure]

In the period of A decrease from 23 April to 6 May (DOY 113-126) , no rain/snow is flagged, so we can hypothesize that the snow removal is due to the melting caused by high temperatures, associated with high DLI values, as explained in the text. The rain flag is on again on 7-8 May (DOY 127-128), when A value rises above 0.70, because of snow precipitation. Finally, A reaches 1 on 13 May (DOY 133), after snowfall.

The pictures taken from the webcam on building 1971 showing the ground around the upward SW and LW irradiances measurement site are very useful in identifying variations on the surface cover. Unfortunately, such pictures are not available for some days. However, the picture of 4 May (DOY 124) shows some patches of snow-free ground, which are responsible for the A decrease. On the contrary, the picture of 14 May (DOY 134), captured after the end of the snowfall, shows a homogeneous snow cover.

[Figure]

[Figure]

Picture of 4 May 2021. The red circle shows the position of the radiometers facing the surface

Picture of 14 May 2023

Figure 10: the relation between T anomalies and albedo isn't obvious: preferably, the upper panel should show the absolute temperature (with indication of 0°C). The same applies to Line 663, where this event is discussed. "11 K higher than unusal" doesn't mean a lot here if no baseline is provided.

We agree with the reviewer, and we added a panel with absolute temperatures, indicating also the x axis corresponding to the melting point (273.15 K). The panel shows that the reduction in A starting by mid April 2021 is triggered by a steep increase in air temperature, which reaches values above the melting point (up to 278.8 K on 26 April, corresponding to an anomaly of 11.4 K). This represents a record for the month, compared to the other years in the 2016-2022 interval.

Similarly, air temperature remains above the melting point for six days by the end of September 2019, with a peak in air temperature of 280 K, corresponding to an anomaly value near 9 K, so again the extraordinarily warm temperature is presumably the factor causing the delay in the snow season.

Some comments have been added in Section 3.2.2 and in the Conclusions.

Figure 14: unnecessary figure, the basic information is included in Figure 16.

Figure 14 has been removed.